# Efficient long-range conduction in cable bacteria through nickel protein wires

Henricus T. S. Boschker [1,2✉], Perran L. M. Cook [3], Lubos Polerecky [4], Raghavendran Thiruvallur Eachambadi [5], Helena Lozano [6], Silvia Hidalgo-Martinez[2], Dmitry Khalenkow[7], Valentina Spampinato[8], Nathalie Claes [9], Paromita Kundu[9], Da Wang[9], Sara Bals [9], Karina K. Sand [10], Francesca Cavezza[11], Tom Hauffman [11], Jesper Tataru Bjerg[2,12,13], Andre G. Skirtach [7], Kamila Kochan[3], Merrilyn McKee[3], Bayden Wood [3], Diana Bedolla [14], Alessandra Gianoncelli [14], Nicole M. J. Geerlings[4], Nani Van Gerven[15,16], Han Remaut[15,16], Jeanine S. Geelhoed [2], Ruben Millan-Solsona [6,17], Laura Fumagalli [18,19], Lars Peter Nielsen [12,13], Alexis Franquet[8], Jean V. Manca [5], Gabriel Gomila[6,17] & Filip J. R. Meysman [1,2✉]

Filamentous cable bacteria display long-range electron transport, generating electrical currents over centimeter distances through a highly ordered network of fibers embedded in their cell envelope. The conductivity of these periplasmic wires is exceptionally high for a biological material, but their chemical structure and underlying electron transport mechanism remain unresolved. Here, we combine high-resolution microscopy, spectroscopy, and chemical imaging on individual cable bacterium filaments to demonstrate that the periplasmic wires consist of a conductive protein core surrounded by an insulating protein shell layer. The core proteins contain a sulfur-ligated nickel cofactor, and conductivity decreases when nickel is oxidized or selectively removed. The involvement of nickel as the active metal in biological conduction is remarkable, and suggests a hitherto unknown form of electron transport that enables efficient conduction in centimeter-long protein structures.

[1] Department of Biotechnology, Delft University of Technology, Delft, The Netherlands. [2] Microbial Systems Technology Excellence Centre, University of Antwerp, Wilrijk, Belgium. [3] School of Chemistry, Monash University, Clayton, Australia. [4] Department of Earth Sciences—Geochemistry, Faculty of Geosciences, Utrecht University, Utrecht, The Netherlands. [5] X-LAB, Faculty of Sciences, Hasselt University, Diepenbeek, Belgium. [6] Nanoscale Bioelectrical Characterization, Institut de Bioenginyeria de Catalunya (IBEC), The Barcelona Institute of Science and Technology, Barcelona, Spain. [7] Department of Biotechnology, University of Ghent, Ghent, Belgium. [8] IMEC, Leuven, Belgium. [9] Electron Microscopy for Materials Research (EMAT), University of Antwerp, Antwerp, Belgium. [10] Department of Chemistry, Nano-Science Center, University of Copenhagen, Copenhagen, Denmark. [11] Research Group Electrochemical and Surface Engineering, Department Materials and Chemistry, Vrije Universiteit Brussel, Brussels, Belgium. [12] Microbiology, Department of Biology, Aarhus University, Aarhus, Denmark. [13] Center for Electromicrobiology, Department of Biology, Aarhus University, Aarhus, Denmark. [14] Elettra-Sincrotrone Trieste S. C.p.A., Trieste, Italy. [15] VIB-VUB Center for Structural Biology, Flanders Institute for Biotechnology (VIB), Brussels, Belgium. [16] Structural Biology Brussels, Vrije Universiteit Brussel, Brussels, Belgium. [17] Departament d'Enginyeria Electrònica i Biomèdica, Universitat de Barcelona, Barcelona, Spain. [18] Department of Physics and Astronomy, University of Manchester, Manchester, UK. [19] National Graphene Institute, University of Manchester, Manchester, UK. ✉email: h.t.s. boschker@tudelft.nl; filip.meysman@uantwerpen.be

Cable bacteria are multicellular microorganisms in the Desulfobulbaceae family that display a unique metabolism, in which electrical currents are channeled along a chain of more than 10,000 cells[1–4]. The observation that electrical currents are transported along centimeter-long filaments[3,4] extends the known length scale of biological electron transport by orders of magnitude, and suggests that biological evolution has resulted in an organic structure that is capable of highly efficient electron transport across centimeter scale distances[4]. Recent studies demonstrate that cable bacteria effectively harbor an internal electrical grid, which displays a unique topology and exceptional electrical properties[5–8]. The cell envelope of cable bacteria contains a distinctive network of parallel fibers (each ~50 nm diameter) that run along the whole length of the filament[5,6]. These fibers are embedded in a joint periplasmic space and remain continuous across cell-to-cell junctions[5]. Direct electrical measurements demonstrate that these periplasmic fibers are the conductive structures[7,8]. Additionally, the cell-to-cell junctions contain a conspicuous cartwheel structure that electrically interconnects the individual fibers to a central node, and in this way, the electrical network becomes redundant and fail-safe[8]. The electrogenic metabolism of cable bacteria necessitates that nano-ampere currents are efficiently conducted over centimeter scale distances through this network[4], and in effect, the periplasmic fibers display extraordinary electrical properties for a biological material[7]. The estimated in vivo current density of ~$10^6$ A m$^{-2}$ is comparable to that of household copper wiring, while the estimated conductivity of the fibers exceeds 20 S cm$^{-1}$ and thus rivals that of doped synthetic conductive polymers[7].

Bio-materials typically have an intrinsically low electrical conductivity, and so the availability of a bio-material with extraordinary electrical properties has great potential for new applications in bio-electronics. This prospect of technological application however requires a deeper understanding of the mechanism of electron transport as well as the structure and composition of the conductive fibers in cable bacteria. Yet at present, these aspects remain highly enigmatic. One important obstacle is that cable bacteria have a highly complex metabolism and life-style, which strongly hampers culturing and biomass collection. The inability to obtain sufficient cable bacteria biomass precludes the implementation of many traditional bulk analytical techniques that could shed light on the chemical composition of the electrical network. Here, we solved this problem by the application of high-resolution microscopy, spectroscopy, and chemical imaging methods to individual filaments of cable bacteria. This approach allows to elucidate the chemical structure and composition of the conductive fibers in cable bacteria.

In this work, we show that the long-range electron transport in cable bacteria is crucially dependent on proteins containing a sulfur-ligated nickel group. This finding sets the conduction mechanism in cable bacteria apart from any other known form of biological electron transport, and demonstrates that efficient conduction is possible through centimeter-long protein structures.

## Results

**Protein fibers on a polysaccharide-rich layer**. Through sequential extraction, a so-called fiber sheath can be isolated from the periplasm of cable bacteria filaments[5], which contains the conductive fibers[7,8]. Previous studies have already elucidated the geometrical configuration of this fiber network[5–8]. Here, we applied high angle annular dark field–scanning transmission electron microscopy (HAADF-STEM) and subsequent tomography, which provided additional details of the fiber sheath architecture (Fig. 1A and Supplementary Movie 1), demonstrating

that the regularly spaced fibers appear to be held together by a basal sheath. The fibers are clearly visible in unstained preparations, thus providing further support that they form electron dense structures[2,5].

Individual fiber sheaths were subjected to various forms of spectroscopy and chemical imaging. Atomic force microscopy–IR spectroscopy (AFM-IR) provided a first insight into the biochemical composition of the fiber sheath (Fig. 1B), and indicated that it mainly consists of protein (N–H/O–H 3300 cm$^{-1}$, Amide I 1643 cm$^{-1}$, Amide II 1562 cm$^{-1}$, Amide III 1290 cm$^{-1}$) and polysaccharide (1202 cm$^{-1}$ and broad feature at 1115–1166 cm$^{-1}$) (band assignment based on[9–11]). The detection of ester groups in the AFM-IR spectra (C=O stretching at 1765 cm$^{-1}$; C=O bending at 1398 cm$^{-1}$) suggested that the polysaccharide contains acidic sugars[9]. A band at 3056 cm$^{-1}$ likely represents aromatic C–H stretching for instance from aromatic amino acids. Cell junctions give similar AFM-IR spectra as central cell areas, although signals are higher in the junctions (Fig. 1B, C), likely due to the presence of the cartwheel structure that interconnects fibers[5] and also seems to be made of protein and polysaccharide. Mapping of the Amide I band gave a relatively even signal across the central cell area with no indication of a fiber structure (Fig. 1C), thus suggesting that there is protein throughout the fiber sheath. A recent genome and proteome study[12] speculates that the periplasmic fibers of cable bacteria could be composed of bundles of pilin protein, as found in the conductive pili of *Geobacter*[13,14]. The position of the Amide I peak at 1643 cm$^{-1}$ suggests however that the protein secondary structure is mainly disordered, and not of the α-helix type[10], which hence speaks against an abundance of α-helix-rich pilin proteins. We estimate that the fibers themselves represent 25% of the total mass of the fiber sheath (see Supplementary Note 3), and so other proteins outside of the fibers can contribute to the Amide I peak. While it is unlikely that pilin protein is present in high concentrations in the fibers, its complete absence requires further confirmation.

Time of flight-secondary ion mass spectrometry (ToF-SIMS) analysis was applied in combination with in situ AFM calibration of the sputtering depth. This enabled us to map the nanometer-scale depth distribution of both organic and inorganic constituents within individual fiber sheaths (additional ToF-SIMS results are provided in Supplementary Note 1). Replicate ToF-SIMS analyses of fiber sheaths in both positive and negative mode yielded a consistent depth distribution of organic fragments (Fig. 1D, E, Supplementary Fig. 1 and Supplementary Data 1 and 2). Initially, high signals were recorded for a variety of amino acid fragments, including all three aromatic amino acids[15,16]. After ~150 s of sputtering, the amino acid-derived signal levelled off, while counts of oxygen-rich fragments, including carbohydrate specific ions ($C_2H_5O_2^+$, $C_3H_3O_2^+$, and $C_3H_5O_2^+$)[17] peaked. In these samples, the fiber sheath forms a flattened hollow cylinder upon the supporting substrate, which has a total thickness of 117 ± 10 nm (superposition of top and bottom cell envelope layers as measured in the middle of a cell; Supplementary Fig. 2). AFM calibration of the sputtering time places the carbohydrate peak at 59 ± 6 nm depth (Supplementary Fig. 2), which matches the middle of the flattened fiber sheath between the top and bottom layers. This suggests that the fiber sheath is made of a protein layer containing the fibers on top of a basal polysaccharide-rich layer.

Together, the HAADF-STEM, AFM-IR, and ToF-SIMS data show that (1) the conductive fibers are positioned in a regular, parallel pattern on the outside of the fiber sheath, (2) that the fibers consist of protein, and (3) that the fibers rest upon a basal sheath rich in polysaccharide. In bacteria with a Gram-negative cell envelope such as cable bacteria, the most likely source of the polysaccharide layer is the periplasmic peptidoglycan layer (see Supplementary Note 1 for further discussion). Cable

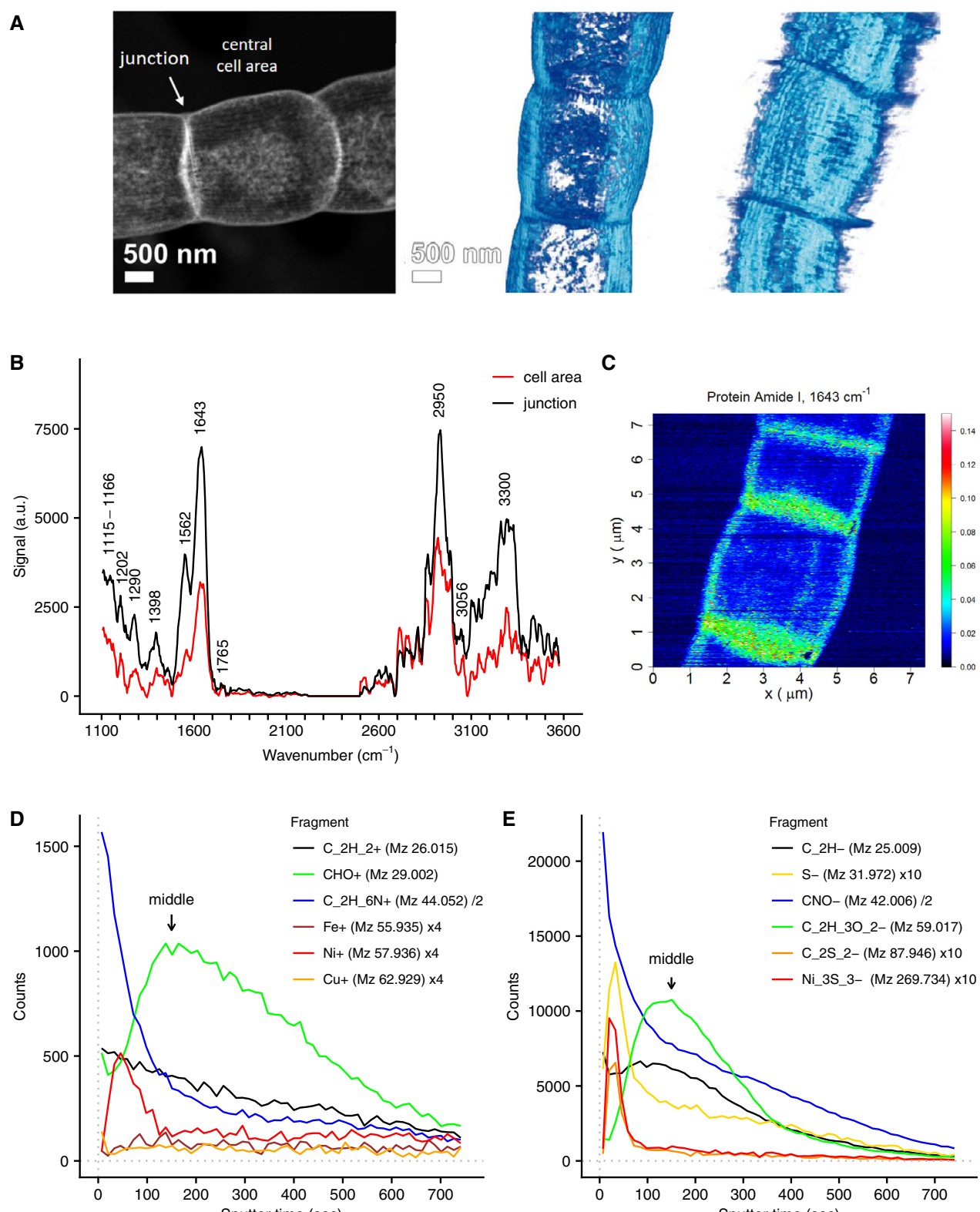

bacteria genomes[12] indeed contain genes encoding for penicillin-binding proteins that perform the final steps in peptidoglycan biosynthesis[18].

**A sulfur-ligated metal group**. To obtain further insight into the composition of the conductive periplasmic fibers, Raman microscopy with different laser wavelengths was applied to both intact cable bacteria and extracted fiber sheaths. Green laser (523 nm) Raman microscopy spectra obtained from living cable bacteria showed the resonance bands of cytochrome heme groups (750, 1129, 1314, and 1586 cm⁻¹) that have been seen previously[3], but additionally revealed two prominent bands within the low-frequency region at 371 and 492 cm⁻¹ (Fig. 2A).

**Fig. 1 The conductive fiber sheath in cable bacteria is composed of a layer of protein on top of an acidic polysaccharide layer. A** STEM-HAADF imaging demonstrates that the fiber sheath is composed of parallel fibers imposed on a basal sheath. One 2D image (left panel) and two 3D tomographic reconstructions are shown. Independent replicas ($N = 2$) showed similar results. **B** AFM-IR spectra of fiber sheaths at cell areas and cell junctions (OPO laser, spectra are background corrected and averaged, cell area $N = 14$, junctions $N = 11$, a.u. is arbitrary units). **C** Fiber sheath AFM-IR mapping of the signal (a.u.) from the 1643 cm$^{-1}$ Amide I protein band (QCL laser, arbitrary units; see Supplementary Fig. 7 for corresponding AFM height and deflection images. Independent replicas ($N = 2$) showed similar results.). Representative ToF-SIMS depth profiles of fiber sheaths obtained in positive (**D**) and negative mode (**E**). A selection of fragments from different compound classes is shown (general organic carbon fragments: $C_2H_2^+$ and $C_2H^-$, protein derived fragments: $C_2H_6N^+$ and $CNO^-$, carbohydrate derived fragments: $CHO^+$ and $C_2H_3O_2^-$ and sulfur and transition metals). See Supplementary Fig. 1 and Supplementary Note 1 for further information. Counts of individual fragments were scaled to improve clarity as indicated in the figure legends. The counts from $Ni_3S_3^-$ are the sum of all $^{58}Ni$ and $^{60}Ni$ isotopologues. Arrows denote the middle of the fiber sheath as calibrated by in situ AFM (59 ± 6 nm, see Supplementary Fig. 2).

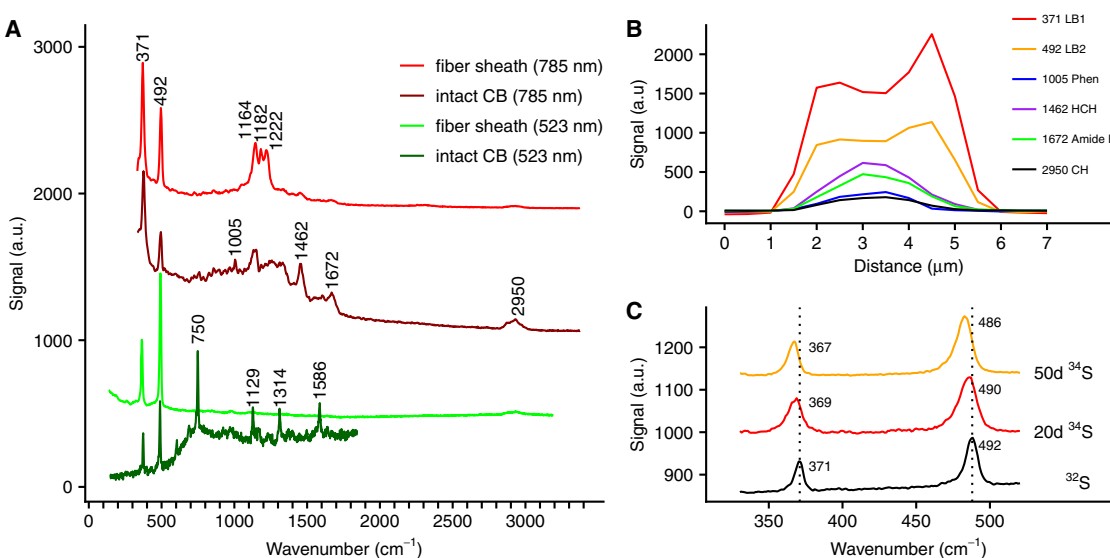

**Fig. 2 Raman spectra of intact cable bacteria and fiber sheaths indicating a sulfur-ligated metal group in the fiber sheath. A** Raman spectra collected with green (523 nm) and NIR (785 nm) lasers. The low-frequency bands at 371 and 492 cm$^{-1}$ suggesting the presence of a metal group, and are present in all spectra. The dark green spectrum is from intact, living cable bacteria (CB) in a gradient slide, while the dark red spectrum is recorded on intact, dried cable bacterium filaments. The light green and light red spectra are from dried fiber sheaths. **B** Variation of the Raman signal (NIR laser) in a transversal section across an intact, dried cable bacterium filament, which was ca. 3 µm wide. The most prominent bands are shown: the two low-frequency bands (371 LB1, 492 LB2), phenylalanine ring-breathing (1005 Phen), CH$_2$-bending (1462 HCH), the protein Amide I band (1672 Amide I) and CH-stretching (2950 CH). **C** Average Raman spectra (green laser) and peak shifts resulting from $^{34}S$ labeling of intact, dried cable bacterium filaments. The two low-frequency bands are shown after 20 and 50 days of incubation and compared to the unlabeled control spectrum.

These two Raman bands remained prominently present when cable bacterium filaments were isolated from the sediment and air-dried. Moreover, the two bands appeared in both thick (~4 µm diameter) and thin (~1 µm diameter) filament morphotypes, as well as in marine and freshwater cable bacteria (Fig. 2A and Supplementary Fig. 3A). This suggests that the two low-frequency bands are a core feature of the cable bacteria clade. The low-frequency position points toward a cofactor that involves a heavy atom, such as a ligated metal group[11].

When air-dried intact cable bacteria were investigated with near infrared (NIR) laser (785 nm) Raman spectroscopy, the two low-frequency bands were again present, but the spectra also showed additional bands (Fig. 2A), characteristic of general biomolecular constituents, such as C–H stretching (2950 cm$^{-1}$), the Amide I peak of protein (1672 cm$^{-1}$), CH$_2$ bending (1462 cm$^{-1}$), and phenylalanine ring-deformation (1005 cm$^{-1}$)[19]. Cross-sectional scans of intact filaments showed a unimodal signal for biomolecular signals generally present in microbial biomass (phenylalanine, C–H bonds, Amide I). In contrast, the signal of the two low-frequency bands showed a bimodal maximum at the filament edges (Fig. 2B), which suggests that the metal moiety is located in the cell envelope, possibly in the periplasmic fiber sheath (Supplementary Fig. 3B). This was confirmed by Raman spectroscopy of fiber sheaths

extracted from intact bacteria (Fig. 2A), which produced simple green laser spectra that only contained the two low-frequency bands and a weak C–H signal at 2950 cm$^{-1}$. Furthermore, these spectra also showed no cytochrome signal[7], thus confirming that the conduction mechanism does not involve cytochromes[7] as seen in *Shewanella* and *Geobacter* nanowires[20–22]. NIR laser Raman spectra of fiber sheaths additionally revealed three conspicuous bands at 1164, 1182, and 1222 cm$^{-1}$ (Fig. 2A), which may originate from single carbon bonds (C–C, C–O, or C–N) associated with the metal group[11], and also showed small bands from protein (Amide I, 1665 cm$^{-1}$) and C–H (1451 and 2950 cm$^{-1}$)[19]. Ratios between the background-corrected peak heights of the two low-frequency bands were similar in all Raman spectra recorded (green laser $R_{492/371} = \sim 2.1$, NIR laser $R_{492/371} = \sim 0.6$), suggesting that both bands originate from a single moiety.

To further examine the origin of the two low-frequency bands, we grew cable bacteria in sediments amended with $^{34}S$ or $^{13}C$ stable isotope tracers and investigated intact air-dried filaments with Raman spectroscopy as before. Labeling with $^{34}S$ did not affect the cytochrome bands as expected, but resulted in a shift in both low-frequency bands toward lower wave numbers (Fig. 2C). This suggests that sulfur is directly involved in both low-frequency bands and the metal group therefore appears to be S-

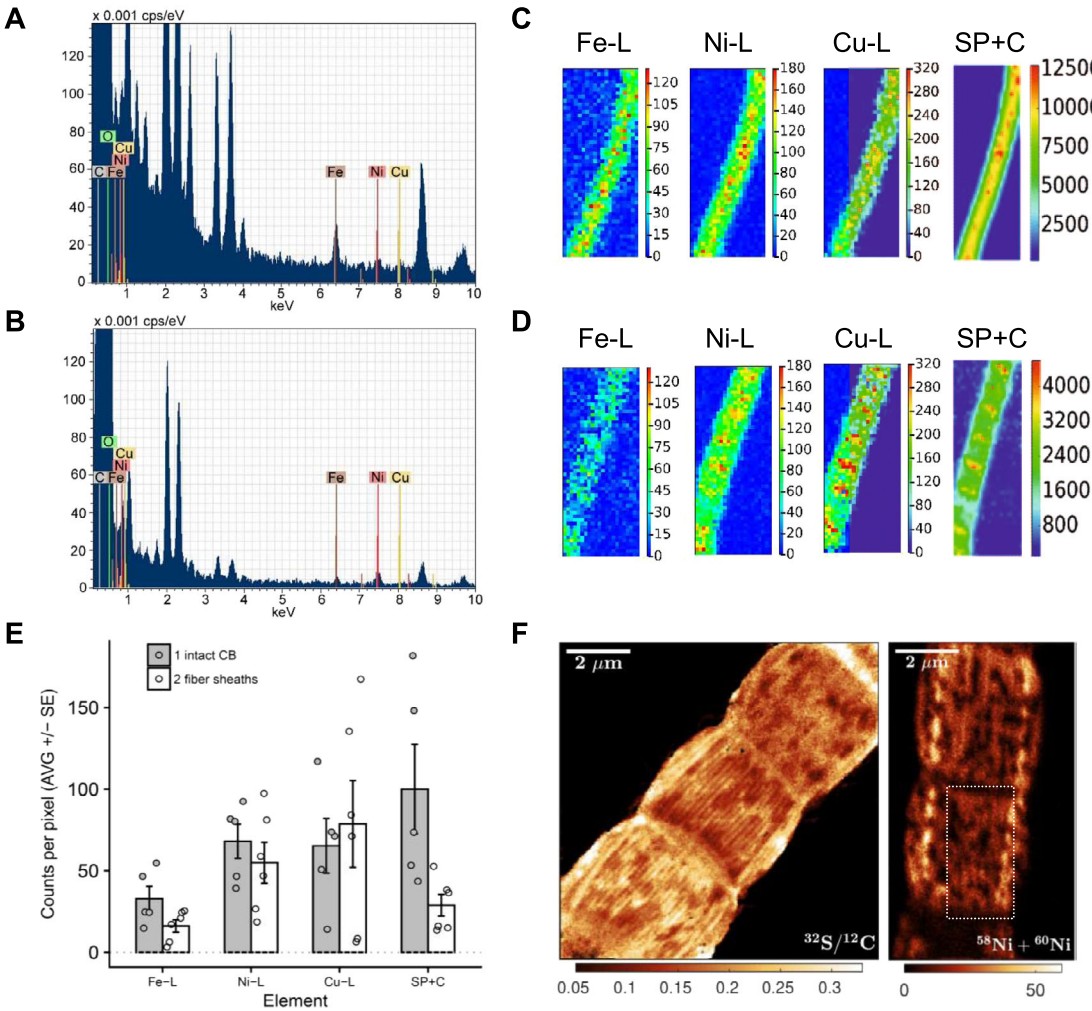

**Fig. 3 Elemental analysis shows that the fibers are Ni and S rich.** Representative STEM-EDX spectra from **A** intact cable bacteria and **B** fiber sheaths shows a detectable Ni signal and lower Fe and Cu levels in the fiber sheath. Elemental compositions are found in Supplementary Table 1. Representative synchrotron LEXRF maps for **C** intact cable bacteria (10 μm × 25 μm) and **D** fiber sheaths (11 μm × 27 μm). SP + C denotes Scatter Peak plus Compton and L denotes low-energy L-band. **E** Average counts per pixel from LEXRF maps showing that Ni is mainly found in the fiber sheath. Shown are the average of the maps ± SE (intact cable bacteria $N = 5$ and fiber sheaths $N = 6$, data are background corrected) and the data points for the individual maps. Data given for the detected transition metals and SP + C. The latter data were scaled to fit into the graph by setting the average of the intact cable bacteria (CB) counts to 100 (original counts 4290 ± 2640). **F** NanoSIMS images of fiber sheaths. Mapping of $^{32}S/^{12}C$ ion count ratio (first 100 planes) shows the sulfur rich fibers. The Ni ($^{58}Ni + ^{60}Ni$) ion count has a lower signal/noise ratio and its mapping (first 50 planes) only shows visible fibers in restricted regions (as indicated by the rectangle). The complete set of NanoSIMS images is given in Supplementary Fig. 5. Independent replicas ($N = 2$) showed similar results.

ligated. Labeling with $^{13}C$ resulted in substantial shifts of the cytochrome bands to lower values, as expected. This also demonstrated that cable bacteria were highly labeled (Supplementary Fig. 4), but nevertheless, the 371 cm$^{-1}$ band showed no response to $^{13}C$ labeling, while the 492 cm$^{-1}$ band only displayed a small shift to lower wave numbers. This suggests that carbon is not directly involved in metal ligation, but could be present further away (e.g., by having carbon atoms adjacent to the sulfur-ligated metal group).

**Fibers are enriched in nickel and sulfur**. To identify the metal in the sulfur-ligated group, we first analyzed the elemental composition of cable bacterium filaments by STEM-energy-dispersive X-ray spectroscopy (EDX) (Fig. 3A, B and Supplementary Table 1). Metals commonly found in metalloproteins were present in intact filaments, but concentrations were low and close to detection limits: Fe (0.033–0.047 Atm%), Ni (0.009 Atm%), and Cu (0.006–0.009 Atm%). After fiber sheath extraction, Ni

(0.016–0.037 Atm%) was selectively enriched by a factor of 2–4 compared to intact filaments. Based on AFM mappings[5], fiber sheaths contain ~4 times less biomass compared to intact filaments, thus explaining selective Ni enrichment. In contrast, Fe was partially removed by a factor of ~2, which is consistent with the loss of cytochrome bands in the Raman spectra after extraction (Fig. 2A). Cu remained equally low. Additional metal analysis by Synchrotron low-energy X-ray fluorescence (LEXRF) (Fig. 3C–E) showed that absolute Ni counts were similar in intact bacteria and fiber sheaths, thus providing further support that Ni is concentrated in the fiber sheath. Fe was again selectively lost during extraction of fiber sheaths, while Cu levels were highly variable and exceeded STEM-EDX values, suggesting that Cu data were affected by contamination during LEXRF analysis.

Together, the STEM-EDX and LEXRF data suggested that Ni was the most likely candidate for the metal contained in the sulfur-ligated group. This hypothesis was further supported by ToF-SIMS analysis of the fiber sheaths (Fig. 1D, a detailed discussion of ToF-SIMS data is given in Supplementary Note 1).

In positive mode the four main Ni isotopes ($^{58}$Ni, $^{60}$Ni, $^{61}$Ni, and $^{62}$Ni) showed a sharp subsurface peak, while the minor isotope $^{64}$Ni showed mass interference (likely from a low amount of $^{64}$Zn). Other transition metals had either low counts (Cu and Fe, Fig. 1D) or were not detectable (Mn, Co, and Mo). Negative mode ToF-SIMS depth profiles showed a subsurface peak of various S-derived anions ($^{32}$S$^-$, $^{34}$S$^-$, SH$^-$, and S$_2{}^-$) at the same position as the Ni peak (Fig. 1E), in agreement with a sulfur-ligated Ni group. The Ni and S peak emerged after 33–46 s of sputtering within the first fiber protein layer (corresponding to $15 \pm 3$ nm of sputtering depth, Supplementary Fig. 2), and was preceded by thin proteinaceous surface layer devoid of Ni.

ToF-SIMS depth profiles from intact cable bacteria also showed subsurface peaks for Ni and S containing fragments (Supplementary Fig. 11). Comparing the relative position of these fragments to fatty-acid fragments derived from the outer and cell membranes suggests that the Ni and S containing fragments originate from the periplasmic space (Supplementary Fig. 11), in agreement with the Ni and S rich nature of the fiber sheath.

High-resolution NanoSIMS analysis confirmed that the conductive fibers were Ni and S rich. S$^-$-ion maps revealed a parallel line pattern (Fig. 3F) with a similar line spacing (~150 nm) between fibers as reported previously (150–200 nm)[5]. Although metal maps generally have a lower signal-to-noise ratio, the $^{58}$Ni$^+$ + $^{60}$Ni$^+$ signal did show an indicative line spacing of ~200 nm in restricted areas (Fig. 3F). Furthermore, the presence of NiS-cluster ions such as Ni$_3$S$_3{}^-$ in the negative mode ToF-SIMS spectra (Fig. 1E and Supplementary Fig. 9) also suggests that Ni and S must be present in close proximity (lateral distance ($XY$) within <0.5 nm, depth ($Z$) within <2 nm), otherwise these NiS-clusters would not form in the ToF-SIMS ion plume[23]. Finally, we detected two organic sulfur fragments (C$_2$S$_2{}^-$ and C$_2$S$_2$H$^-$) that were specifically associated with the Ni$_3$S$_3{}^-$ peak (Fig. 1E and Supplementary Fig. 1). Fiber sheaths display an exceptional chemical resistance, as the fiber extraction protocol uses relatively strong chemical agents (SDS and EDTA), but the fiber network remains structurally intact (Fig. 1A and Supplementary Movie 1, and ref. [5]) and functionally intact (conductivity is retained after SDS/EDTA extraction, Fig. 4B, D and ref. [7]). This chemical resistance could be aided by protein disulfide bridges, which could be abundant as suggested by the high S content (Fig. 4F). The two observed organic S fragments (C$_2$S$_2{}^-$ and C$_2$S$_2$H$^-$) could hence come from the disulfide bridges. Alternatively, these two organic fragments could derive from the Ni ligating group. Combined, our results demonstrate that the individual fibers are rich in Ni and S, and that Ni represents the metal in the sulfur-ligated group as detected by Raman analysis.

**The Ni/S group and long-distance electron transport**. To verify whether the Ni/S group truly plays a role in long-distance electron transport, we studied the effect of redox state on Raman signals and conductance. Chemically reducing the fiber sheath with K$_4$Fe$^{II}$(CN)$_6$ resulted in a small increase in the green laser Raman signal of the two low-frequency bands, while oxidizing the fiber sheath with K$_3$Fe$^{III}$(CN)$_6$ almost completely removed these signals (Fig. 4A). Such oxidation state dependent Raman behavior is commonly observed in metalloproteins, such as cytochromes and [FeNi]-hydrogenases, where only one state shows a high resonance Raman signal[3,24]. Subsequent reduction with K$_4$Fe$^{II}$(CN)$_6$ restored the Raman signal, suggesting that the Ni/S group is a reversible redox group. Intriguingly, the conductance of the fiber sheath was also reversibly affected by the redox state of the Ni/S group (Fig. 4B). Reduced fiber sheaths showed a $2.1 \pm 0.5$ ($N = 11$) higher conductance than oxidized fiber sheaths (independent of the direction of the oxidation/reduction step). Note

that the Ni group is possibly not completely reduced, as measurements were done in air, and so the effect of oxidation state on conductance may actually be larger. The decrease of conductance upon oxidation is consistent with previous observations, showing that the conductance of the fiber sheath decreases in ambient air[7]. This suggests that the Ni/S group is oxidized upon exposure to oxygen, inducing a loss of conductivity.

Additional experiments, in which Ni was partially removed from the fiber sheath through extraction with high EDTA concentrations, provided further support that the Ni/S group plays a crucial role in electron transport. EDTA is a known mobilizing agent for metals in biostructures. Raman signals from the SDS only treatment and the standard SDS + EDTA extraction were similar (Fig. 4C) suggesting that an EDTA treatment at low (1 mM) concentrations did not significantly affect the Ni/S group. However, extraction with 50 mM EDTA caused a decreased the Raman signal by 45% suggesting that Ni was selectively removed (Fig. 4C). Fiber structure remained intact (Supplementary Fig. 6), but we can however not fully exclude that other metals (e.g., Ca and Mg) were also partially removed from the fiber structures by the high (50 mM) EDTA treatment, which may have affected secondary and tertiary protein structures to some extent. At the same time, the 50 mM EDTA treatment reduced the conduction on average by 62% (Fig. 4D), which hence could suggest that the Ni/S group plays a key role in maintaining high rates of long-distance electron transport in cable bacteria.

**The core-shell model of a conductive fiber**. By combining and integrating the various types of compositional data collected, we can construct a chemical model of the conductive fiber sheaths in cable bacteria (Fig. 5). The fibers are found on the outside of the fiber sheath and primarily consist of protein (Fig. 1). On the cytoplasmic side of the fiber sheath, the fibers are embedded in or attached to a polysaccharide-rich layer (Fig. 1). This polysaccharide layer holds the fibers together and possibly adds tensile strength to the fiber sheath, which can withstand high pulling forces during filament extraction. As argued before, this polysaccharide-rich layer is most likely the peptidoglycan layer as commonly found in bacteria with a Gram-negative cell envelope such as cable bacteria[18].

ToF-SIMS analysis (Fig. 1D, E) further suggests that the fibers themselves are composed of two distinct regions. The central core of the fiber contains protein material that is rich in Ni, while it is also surrounded by a thin layer of Ni deficient protein (Fig. 5A). This core/shell model is consistent with recent conductive AFM investigations of fiber sheaths, which reveal that fibers only display electrical conductivity when a non-conductive surface layer is first etched away[8]. The cross-sectional structure of the fibers therefore resembles a standard household electrical wire, with a conductive core surrounded by an electrically insulating shell layer. We speculate that this insulating layer prohibits that electrons go astray during long-range transport, thus avoiding radical formation and damage to the surrounding cell environment. However, the insulating outer layer also provokes questions on how electrons generated during electrogenic sulfur oxidation are uploaded onto or downloaded from the conductive core, i.e., how these electrons pass through the insulating layer of the fibers to reach the inner conductive core. Cable bacteria must contain a mechanism for this electron transport to and from the conductive fiber core, which may involve periplasmic cytochromes[12] that are present in intact bacteria but are removed during fiber sheath extraction (Fig. 2A and ref. [7]).

Our fiber core/shell model was independently verified by scanning dielectric microscopy (SDM), which enables AFM-based electrostatic force detection[25,26]. We analyzed a single,

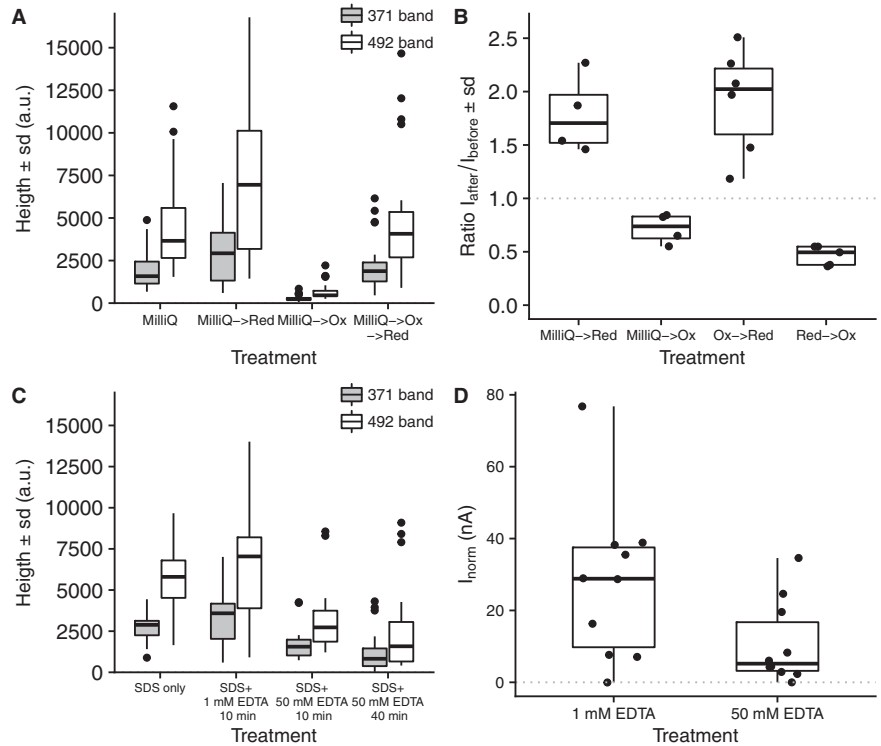

**Fig. 4 Redox and Ni-removal experiments suggest that the Ni/S group plays a role in electron conduction. A** The effect of oxidation and reduction on green laser Raman signals from the sulfur-ligated Ni group. The MilliQ → Red treatment ($N = 30$) was significantly higher than the MilliQ treatment ($N = 28$, $p = 0.03$) and the MilliQ → Ox treatment ($N = 23$) was significantly lower than all other treatments ($p < 4 \times 10^{-9}$). The MilliQ → Ox → Red treatment ($N = 33$) was not significantly different from the MilliQ ($p = 0.7$) and the MilliQ → Red ($p = 0.07$) treatments The ratio between the 371 and 492 cm$^{-1}$ Raman bands was not affected by oxidation or reduction. **B** The effect of oxidation and reduction treatments on the conductance of individual fiber sheaths. The ratio of the electrical current (I) through the fiber sheath is plotted before and after treatment. The effect in all treatment pairs was significant (MilliQ → Red $p = 0.01$ ($N = 4$); MilliQ → Ox $p = 0.01$ ($N = 4$); Ox → Red $p = 0.001$ ($N = 6$); Red → Ox $p = 0.004$ ($N = 5$), all tested against no effect Ratio = 1). **C** The effect of EDTA with on green laser Raman signals from the sulfur-ligated Ni group. The decrease in Raman signal between the standard protocol ($N = 13$) and both high EDTA treatments was significant (10 min. 50 mM EDTA $p = 0.04$ ($N = 15$); 40 min. 50 mM EDTA $p = 0.03$ ($N = 22$)). The SDS only treatment ($N = 15$) was not different from the standard protocol ($p = 0.7$). **D** The effect of high EDTA extraction on normalized conduction of individual fiber sheaths ($I_{norm}$: electrical current normalized to filament length 0.3 mm and bias 0.1 V). Fibers sheaths were extracted with the standard protocol (1% SDS + 1 mM EDTA 10 min) and high EDTA treatment (1% SDS + 50 mM EDTA 10 min). The decrease in $I_{norm}$ between the standard protocol and the high EDTA treatment was significant ($p = 0.041$, $N = 10$). All replicas are for independent cells (Raman data) or fiber sheaths filaments (conduction data). The two-sided Mann–Whitney–Wilcoxon test was used to test for significance using the R-function "wilcox.test". The center line of the box–whisker plots represents the median. The lower and upper box limits represent the 25% and 75% quantiles, respectively. The whiskers extend to the data range. Outliers are indicated in **A** and **C**; all data point are given in **B** and **D**.

isolated fiber that had separated from a fiber sheath (Fig. 5B), and interpreted the modulus and phase (Fig. 5C, D) of the 2ω-electric force harmonic with a computational finite-element model of a flattened cylindrical fiber (right insert in Fig. 5C; height = 41 nm; width = 87 nm, obtained from the deconvoluted topographic image in the insert of Fig. 5B). When this model assumed that the fiber was conductive ($\sigma_c = 20$ S cm$^{-1}$ as determined in[7]) and homogeneous (only core, no shell), it could not fit both the modulus and phase data of the electric force (see Supplementary Note 2 and Supplementary Fig. 13 for details). Alternatively, when we assumed the fiber was homogenous and non-conductive ($\sigma_c = 0$ S cm$^{-1}$), this resulted in an anomalously high relative permittivity $\varepsilon_r = 11 \pm 3$, implying that the dielectric response of the fiber material would substantially exceed the typical values for common proteins ($\varepsilon_r = 3$–5)[25–28], and would even surpass that of nucleic acids ($\varepsilon_r = \sim 8$)[25,26]. This is not congruent with our AFM-IR and ToF-SIMS data, which demonstrate that the fibers are made of protein. However, when we parameterized the fiber model to include a conductive protein core ($\varepsilon_c = 3$; $\sigma_c = 20$ S cm$^{-1}$) surrounded by a non-conductive protein shell ($\varepsilon_s = 3$ conductivity $\sigma_s = 0$ S cm$^{-1}$), we could fit

both the modulus and phase data of the electric force, arriving at a shell thickness $d = 12 \pm 2$ nm (red dashed lines in Fig. 5C, D). The SDM data therefore add further support to the proposed core/shell model (see Supplementary Note 2 for a full description of SDM results and models tested).

## Discussion

Our results indicate that conduction in cable bacteria occurs through proteins with Ni-dependent cofactors. The observation that Ni plays a crucial role in long-range biological conduction is remarkable, as biological electron transport typically involves Fe and Cu metalloproteins[29], though not enzymes with Ni centers. Nickel acts as a catalytic center in only nine enzymes, which are mostly involved in the metabolism of gases[30–32], but not in electron transport.

The sulfur-ligated Ni group in the periplasmic fibers has a well-defined Raman signature, which does not resemble that of any of the known sulfur-ligated nickel enzymes[24,33]. Clearly, we are dealing with a new type of Ni cofactor, and our data provide a first insight into the structure of this sulfur-ligated Ni-dependent

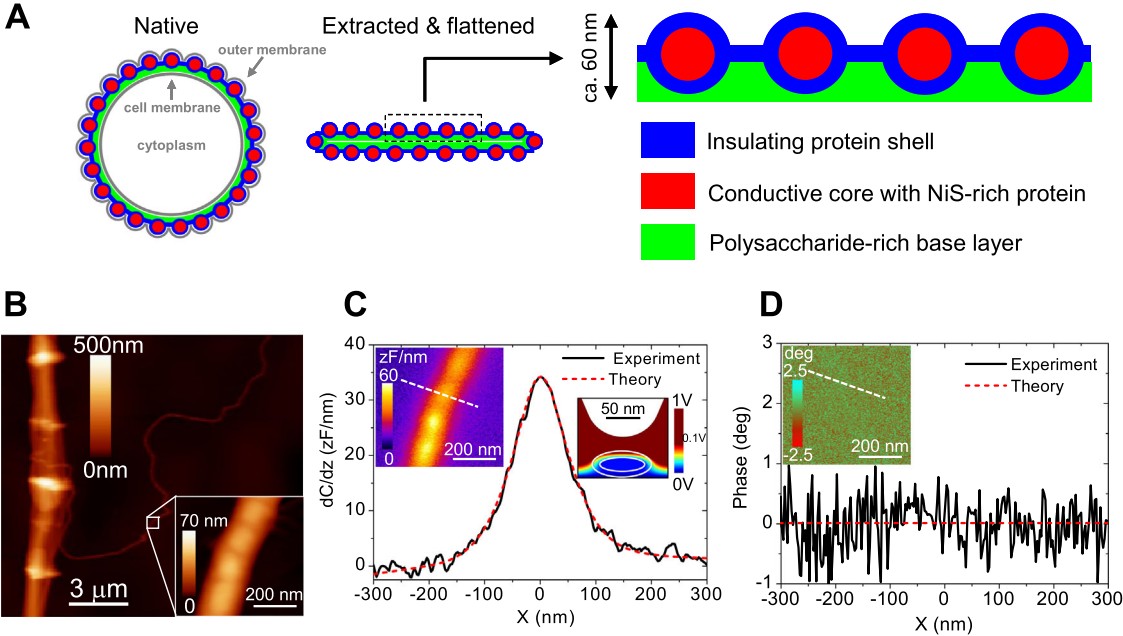

**Fig. 5 Fiber sheath model and electrostatic properties. A** Compositional model of the conductive fiber sheath in cable bacteria based on the present findings. Cross-sections through a filament in the middle of a cell are drawn and the number of fibers has been reduced for clarity—a 4 µm diameter cable bacterium has typically ~60 fibers[5]. In its native state (right panel), the fiber sheath is embedded periplasm between the cell and outer membrane and adopts a circular shape. After extraction, which removes the membranes and most of the cytoplasm and after drying upon a surface for analysis, the fiber sheath flattens, leading to two mirrored sheaths on top of each other (middle panel). The enlargement shows a section of the top sheath, which is the sample section probed by ToF-SIMS depth profiles and NanoSIMS images. Fibers are made of protein with a conductive Ni/S rich core and a non-conductive outer shell, and are embedded in a basal layer enriched in polysaccharide. **B** Topographic AFM image of a fiber sheath with a single isolated fiber detaching. The insert shows a detailed AFM image of this single fiber. **C** SDM amplitude image (right insert) and cross-sectional profile. **D** Corresponding SDM phase image (insert) and cross-sectional profile. Constant height ($z = 66$ nm) cross-section profiles are measured along the dashed lines shown in the left inserts. The red dotted lines in **C** and **D** represent model fits assuming the a fiber has a conductive core and an insulating outer shell. The right insert in **C** shows a vertical cross-section of the electric potential distribution as predicted by the model. Model parameters: shell thickness, $d = 12$ nm; fiber height, $h = 42$ nm; fiber width $w = 87$ nm; relative dielectric constants of the shell and core, $\varepsilon_s = \omega \varepsilon_c = 3$; conductivity of the shell $\sigma_s = 0$ S/cm (insulating); conductivity of the core $\sigma_c = 20$ S/cm[7] (see Supplementary Note 2 for treatment of SDM results and models tested). SDM analysis on a single fiber is available only from one samples as this is a rare event, but results from a double fiber and fiber sheaths are in agreement (see Supplementary Note 2).

cofactor. The low-frequency band at 371 cm$^{-1}$ (Fig. 2) is most likely due to Ni–S bond stretching, and bands at similar wave numbers are found in S-ligated Ni-metalloproteins[24,33] and NiS minerals[34–36]. However, these spectra typically show additional smaller bands from other Ni–S vibrational modes[24,33–36], which are absent in the fiber sheath spectra (Fig. 2). Our stable isotope labeling data suggest that the second low-frequency band at 492 cm$^{-1}$ also involves sulfur and maybe indirectly carbon. While a similar band is observed in NiS$_2$ mineral spectra, these spectra also show additional bands around 270 cm$^{[-1}$ [34,36] that are not seen in the fiber sheath spectra, leaving the alternative possibility that the 492 band derives from S–S stretching[11]. Finally, both the green and NIR Raman spectra show some resemblance to Ni-bis-dithiolene ligands[37,38]. The 492 band then would come from ring-breathing of the aromatic ring containing the dithiolene, and the middle bands in the NIR Raman spectrum (at 1164, 1182, and 1222 cm$^{-1}$, Fig. 2A) could originate from C–C or C–N stretching in the aromatic ring[37,38]. Also, the two organic sulfur fragments associated with the conductive core as detected by ToF-SIMS could possibly be derived from a dithiolene ligand (C$_2$S$_2^+$ could be S–C=C–S$^+$, for instance). Future studies should better resolve the molecular structure and Ni center coordination of this novel cofactor, and clarify its role in electron transport.

While protein is generally considered to be an electrical insulator, recent work demonstrates that electrical currents can propagate efficiently through nanometer-thick protein films

sandwiched between electrodes[39,40], as well as through micrometer-long protein appendages of metal-reducing bacteria from the genera *Shewanella* and *Geobacter*[20–22]. Our data suggest that the periplasmic fibers in cable bacteria consist of protein, with an outer layer of Ni deficient protein that is electrically insulating, and a central core of Ni-rich protein that is electrically conductive. Our results thus extend this known length scale of protein conduction from micrometers to centimeters. The periplasmic fibers in cable bacteria are continuous across cell–cell junctions, and hence, they provide one uninterrupted pathway for electron transport along these centimeter-long bacterial filaments[5].

While conduction in *Shewanella* nanowires occurs through Fe-containing cytochromes[20], up until recently, *Geobacter* nanowires were exclusively thought to consist of metal-free pilin protein, where pi–pi stacking of aromatic amino acids provides a conductive pathway[13]. This view of electron conduction in *Geobacter* nanowires has substantially changed in recent years, as high-resolution molecular analysis has revealed that these nanowires can be composed of stacks of OmcS or OmcZ multiheme cytochromes[21,22,41]. Together, our data suggest a mechanism of long-range electron transport in cable bacteria that is hitherto unknown to science. The involvement of Ni in conduction and the lack of cytochromes in the periplasmic fibers of cable bacteria imply a different mechanism of electron transport than in *Shewanella* and *Geobacter* nanowires.

As electron conduction in cable bacteria needs to spans centimeters, the electron transport needs to be highly efficient, thus requiring organic structures with high conductivity. When it was discovered that the periplasmic fibers in cable bacteria were the current bearing structures, it was already noted that mean fiber conductivities were high[7], providing typical values of ~10 S cm$^{-1}$ with a maximum up to 79 S cm$^{-1}$, which are comparable to those of highly doped organic semiconductors. The core-shell model proposed here suggests that only the conductive core (26 nm diameter) channels the current, thus condensing the electron flow to within 27% of the total cross-sectional area of the 50 nm diameter fiber. This implies that the Ni-containing protein core has a typical conductivity of ~37 S cm$^{-1}$, with a maximum up to 292 S cm$^{-1}$. The conduction data for the standard SDS-EDTA extracted fiber sheaths in Fig. 4D lead to similar conductivities with an average of 26 S cm$^{-1}$ and a maximum of 73 S cm$^{-1}$. These conductivities are substantially higher than measured for the cytochrome based nanowires in *Shewanella* (1 S cm$^{-1}$ [20]) and the OmcS type nanowires in *Geobacter* (0.02–0.04 S cm$^{-1}$ [21,22]), but comparable to the more conductive OmcZ cytochrome nanowires from *Geobacter* (4–30 S cm$^{-1}$ at pH 7 to 373 S cm$^{-1}$ at pH 2[22]). A clear challenge is to establish a solid physical and molecular foundation for these high protein conductivities of cable bacteria fiber cores.

The exact mechanism of electron transport in the periplasmic fibers of cable bacteria remains unclear, but our results indicate that the novel Ni cofactor is an essential component. Electron transport in metalloproteins critically depends on the distance between metal centers[42], and assuming an homogeneous distribution of Ni atoms throughout the conductive fiber core (see Supplementary Note 3), the distance between Ni atoms is 1.4–1.9 nm, which leads to a characteristic electron tunneling time of 50–90 ns. This interatomic Ni distance is in the range commonly found for metal centers in metalloproteins, but the electron tunneling times are on the fast end of what has been observed for single electron transfers[42]. In comparison, the interatomic distance between Fe atoms in OmcZ cytochrome based nanowires from *Geobacter* is somewhat smaller (on average 1.2 nm) and tunneling times are only ~5 ns (both calculated from[22]). It has been argued that the high conductivity of OmcZ nanowires is primarily due to the enhanced pi/pi stacking of closely arranged heme groups[22]. The fast electron tunneling times of 50–90 ns suggests that the conductive fiber core must possess a supramolecular organization that supports highly efficient electron transport between Ni centers. This could involve particular configurations of aromatic amino acids, or maybe an unknown prosthetic group involved in Ni-ligation. Future studies should further elucidate the genetic and molecular basis of the highly efficient long-range electron transport in cable bacteria.

## Methods

**Cultivation and sample preparation.** Cable bacteria were cultured in natural sediments which were sieved, homogenized, repacked into PVC core liner tubes (diameter 40 mm) and incubated in aerated artificial seawater (ASW) at in situ salinity (see[43]). Enrichment cultures with sediment collected from a salt marsh creek bed (Rattekaai, The Netherlands) consistently developed thick (~4 µm diameter) filaments that were used for majority of the experiments (unless stated otherwise). To verify whether Raman spectra were similar for different groups of cable bacteria, filaments were retrieved from enrichment cultures using sediments from other environments: Mokbaai (The Netherlands), marine intertidal sediment providing thick filaments (~4 µm); Aarhus university pond (Denmark), a freshwater site with thin filaments (~1 µm); Yarra River, Australia, brackish estuarine sediment, generating thin filaments (~1 µm) that were also used for stable isotope labeling.

Under a stereo microscope, cable bacterium filaments were gently pulled from the top layer of the sediment with glass hooks that were custom made from Pasteur pipettes. Subsequently, cable bacterium filaments were cleaned and washed by transferring them at least six times between droplets (~ 20 µl) of MilliQ water on a microscope cover slip, thus providing "intact cable bacteria filaments". Fiber sheath

samples were obtained by extraction of intact filaments in a droplet (~20 µl) of 1% (w/w) sodium dodecyl sulfate (SDS) for 10 min followed by six MilliQ washes. Subsequently, filaments were incubated for 10 min in a droplet (~20 µl) of 1 mM sodium ethylenediaminetetraacetate (EDTA), pH 8, and again six times washed in MilliQ[5].

**Confocal Raman microscopy.** Raman spectra were recorded on several confocal microscopy systems equipped with different lasers: Renishaw InVia (532 nm green laser, 30 mW), Horiba LabRAM (532 nm green laser, 500 mW), and WITec confocal CRM alpha 300 Raman microscope (30 mW 532 nm green laser and 180 mW 785 nm NIR laser). Intact cable bacterium filaments and fiber sheaths were deposited on various substrates (glass microscope slides, cover slips, Al-coated Petri dishes) for the green laser Raman analysis and Raman grade CaF$_2$ covers for combined green laser and NIR laser Raman analysis. In addition, we also recorded green laser Raman spectra on living cable bacteria in sediment gradient slides[3]. Raman spectra were subjected to cosmic spike removal (CRR), averaged, and baseline corrected (NIR laser spectra only, using the Asymmetric Least Squares method from the "Baseline" package in R[44]). The background spectrum—as recorded next to the filaments—was then subtracted.

**Atomic force microscopy–infrared spectroscopy (AFM-IR).** Fiber sheaths were deposited on gold-coated silicon wafers, and AFM-IR spectra were recorded on a NanoIR2 instrument (Anasys). An OPO laser covering the IR range from 900 to 2235 and 2500 to 3600 cm$^{-1}$ (1.3 to 4.7% laser power) was used to record IR spectra. A QCL laser (1500–1700 cm$^{-1}$ at 10% laser power) was used to map the Amide I band maximum at 1642 cm$^{-1}$. We recorded AFM images of samples before and after recording the IR spectra to ensure we did not damage the fiber sheath during the collection of spectra. Contact mode AFM was used throughout with a spring constant $k$ ~0.07–0.4 N m$^{-1}$. Tips were supplied by Anasys and are custom made for the NanoIR2. Spectra from both the central cell area and the cell junctions were recorded. Spectra were smoothened with a Savitzky–Golay filter, averaged and the background as recorded next to the filaments was subtracted.

**Scanning dielectric microscopy (SDM).** SDM was performed following the methods developed in[25]. Briefly, an AC voltage of amplitude 4 V and frequency 2 kHz was applied between a conductive AFM probe (PtSi-CONT from Nanosensors Gmbh with equivalent spring constant $k$ ~0.2 N m$^{-1}$ and resonance frequency $f_0$ ~13 kHz) of a Cypher S AFM (Oxford Instruments, UK). Fiber sheaths were deposited on highly oriented pyrolytic graphite (HOPG) substrate (µMash Inc.). The amplitude and phase of the $2\omega$ harmonic of the cantilever oscillation were recorded in constant height mode at 0% RH (N$_2$ flow) in SNAP mode. The oscillation amplitude was converted to a capacitance gradient[25]. The experimental noise was ~0.7 zF nm$^{-1}$ for the amplitude and ~0.4° for the phase[25,28,45].

For quantitative analysis through finite-element modeling, we described a periplasmic fiber using the elliptic cylinder model as developed in[28] to which we added a core-shell structure with a conductive core and a surrounding shell of thickness $d$. The fiber dimensions (major and minor semi-axis) were extracted from the deconvoluted topographic images. The accuracy of the elliptic cylinder model was assessed against a more realistic model that includes the small (±5 nm) surface topographic variations of the fiber. This analysis supports the assumption that electrical properties are homogeneous along the longitudinal axis of the fibers (see Supplementary Note 2). The core and shell were assumed to have conductivities $\sigma_c$ and $\sigma_s$, and relative permittivities $\varepsilon_c$ and $\varepsilon_s$, respectively. The AFM tip was modeled as a truncated cone ending with a spherical apex[25]. The radius and half cone angle were calibrated from approach curves measured on a bare part of the HOPG substrate[25]. The remaining tip parameters were left to their nominal values (cone height 12.5 µm, disc cantilever thickness 3 µm, cantilever disc radius 3 µm). The macroscopic cantilever contribution was modeled by a constant offset[28]. In the model, a sinusoidal voltage of frequency $\omega$ and amplitude $v_{ac}$ is applied between the tip and the metallic substrate. The force acting on the tip (modulus and phase) was calculated by solving the built-in electric currents model, which employs the AC/DC electrostatic module of COMSOL Multiphysics 5.3 with software routines written in Matlab (Mathworks Inc.), as previously described[25,28,45] (see also Supplementary Note 2).

**Scanning transmission electron microscopy–energy-dispersive X-ray spectroscopy (STEM-EDX).** Intact cable bacterium filaments or fiber sheaths were deposited on a Formvar-coated Au grid or Quantifoil Cu grid and excess water was removed by filter paper. For STEM-EDX, the grid with the filaments was first dried, whereas for the 3D tomography, the grid was immediately plunged in liquid ethane using a Vitrobot Mark 2 plunge freezer (Thermo Fischer Scientific). Afterwards, grids were transferred to liquid nitrogen (~77 K) and mounted into a Gatan cryo holder. Finally, the sample was freeze dried over a period of 8 h in vacuum at 77 K and subsequently allowed to heat to room temperature[46]. With this sample preparation method, the collapse of the fiber sheath, which happens during conventional drying on a TEM grid, could be prevented. Samples were characterized by HAADF-STEM, electron tomography and energy dispersive X-ray spectroscopy (EDX) using a Tecnai Osiris microscope (Thermo Fischer Scientific) operated at 200 kV. For electron tomography, projection images were acquired over a tilt range

of ±50° with an increment of 2° and reconstructed using a simultaneous iterative reconstruction technique implemented in the Astra toolbox[47]. The EDX measurements were carried out using a ChemiSTEM system and analyzed using the Bruker ESPRIT software.

**Synchrotron low-energy X-ray microscopy and X-ray fluorescence (XRM and LEXRF)**. Intact cable bacteria and extracted fiber sheaths were deposited on Formvar-coated Au TEM grids and dried. X-ray microscopy and fluorescence experiments were carried out at the TwinMic Beamline[48] in the Elettra Sincrotrone (Trieste, Italy). For the experiments, TwinMic was operated in scanning mode under a microprobe delivered by an Au zone plate diffractive optic with 600 μm diameter and an outermost zone of 50 nm. In this modality, a fast readout CCD camera produces absorption and phase contrast images, thus providing morphological information on the scanned areas. Simultaneously, LEXRF spectra can be acquired by eight silicon drift detectors places in front of the sample, providing the elemental distribution of excited elements. The excitation energy was chosen accordingly to the elements of interest. Here, an excitation energy of 2 keV was used to have optimal excitation of Si, Al, Mg, Fe, Ni, and Cu. The beam spot size was 570 nm. Firstly, samples were examined with a light microscope to select the regions of interest, then placed inside the TwinMic vacuum chamber operating at $10^{-6}$ mbar pressure. The acquisition time was typically 20 s per pixel. LEXRF maps were obtained by processing the LEXRF spectra with the PyMCA multiplatform software[49].

**Secondary ion mass spectroscopy (SIMS)**. The elemental composition of the fiber sheaths was assessed by nano-scale secondary ion mass spectrometry (NanoSIMS 50 l, Cameca) operated at Utrecht University. Upon extraction, fiber sheaths were transferred onto a polycarbonate filter (pore size 0.2 μm, Isopore, Millipore, the Netherlands) pre-coated with a 5-nm thick gold layer and air-dried in a desiccator for at least 24 h. Fields of view selected with scanning electron microscopy were first implanted with low-energy Cs$^+$ ions (<500 eV) to increase the secondary ion yield without sputtering the material of the sample. Subsequently, the high-energy primary Cs$^+$-ion beam (16 keV) was rastered over the sample surface while detecting secondary ions $^{12}C^-$, $^{16}O^-$, $^{12}C^{14}N^-$, $^{31}P^-$, and $^{32}S^-$. To ensure high lateral (~50 nm) and depth resolution, the primary ion current of 0.5 pA and dwelling time of 1 ms/pixel were used, and the same area (10 × 10 microns, resolution 256 × 256 pixels) was imaged multiple times (600 planes total). The magnetic field strength and exact positions of the electron multiplier detectors were adjusted using an element standard (SPI Supplies, 02757-AB 59 Metals and Minerals Standard). The obtained image data was processed with the Look@NanoSIMS software[50].

A similar approach was used for the detection of positive secondary ions $^{56}Fe^+$, $^{58}Ni^+$, $^{60}Ni^+$, $^{63}Cu^+$, and $^{66}Zn^+$ by NanoSIMS, with the following exceptions. Sputtering was done with the primary O$^-$-ion beam (2 pA, beam size ~100 nm) generated by the radio-frequency plasma source. No low-energy implantation was done with the O$^-$-ion beam. Due to low signals, dwell time was increased to 5 ms per pixel to ensure reliable alignment of the imaged planes; a total of 168 frames were collected. When detecting zinc, isotope $^{66}Zn$ was selected because the position of the detector measuring $^{63}Cu$ was too close to the position required for the measurement of the more abundant isotope $^{64}Zn$.

ToF-SIMS (TOF.SIMS 5, IONTOF, Germany) analysis was performed on intact cable bacterium filaments and fiber sheaths to obtain high mass resolution spectra and detailed depth profiles of both organic fragments and metals. A bundle of well washed filaments was deposited on gold-coated silicon wafers. Using incident light microscopy, spots with free lying filaments were identified for ToF-SIMS analysis. The ToF-SIMS was operated in mass spectrometry mode with a Bi$_3^+$ analytical beam (energy 30 keV, current ~0.35 pA, 100 × 100 μm² area, 256 × 256 pixels) and an Ar$_{4000}^+$ gas cluster ion beam (energy 10 keV, current ~1 pA, 400 × 400 μm² area) was interlaced to obtain depth profiles. The SurfaceLab software (IONTOF, Germany) was used for data analysis. Mass spectra were internally calibrated with C$^+$, CH$_3^+$, C$_2$H$_3^+$, C$_4$H$_5^+$, and Au$^+$ for positive mode and CH$^-$, C$_2$H$^-$, C$_3$H$^-$, C$_4$H$^-$, and Au$^-$ for negative mode spectra and peaks were identified based on exact mass and isotopic composition for metal ions in combination with the Hybrid-SIMS analysis (see below). Lateral regions-of-interest were selected for the filaments and potential spots of sediment clay particles or minerals (high signals of S$^-$, P$^-$, Fe$^+$, Ca$^+$, P$^+$, Al$^+$, and/or Si$^+$) or salt (high signals of Na$^+$, K$^+$, Cl$^-$) were excluded.

Sputtering depths were calibrated with an in situ AFM that was operated in contact mode. To constrain analysis time, a subarea of either 50 × 50 μm² or 20 × 20 μm² (both 256 × 256 pixels, 8 μm/s) of the ToF-SIMS analytical area was imaged by AFM. AFM images before analysis and just after the Ni peak or the carbohydrate peak were recorded and aligned and subtracted in the Gwyddion software to obtain average sputtering depths for the fiber sheaths (see Supplementary Fig. 10 for an example of the data treatment).

Additionally, we analyzed fiber sheaths with a Hybrid-SIMS (IONTOF, Germany) operated with an Ar$_{4000}^+$ cluster analytical beam and equipped with an Orbitrap mass spectrometer. The advantage of the Orbitrap analyser is the higher mass resolution (~240,000) compared to standard ToF-SIMS (~6000 for our samples), and therefore, it was used to identify fragment ions in both positive and in negative modes and to check peak purity. Disadvantages of the Orbitrap are that masses below ca. 50 amu cannot be assessed, which is a minor issue as ToF

identification is generally sufficient here, and the detection limit is poorer because of a ca. 200 count background.

To assist with the interpretation of the ToF-SIMS spectra and the identification of the polysaccharide layer, we analyzed additional samples as reference: bovine serum albumin (BSA), starch from potato, pectin from apple, peptidoglycan from *Methanobacterium* (all from Sigma-Aldrich) and a 1:5 mixture of freshly precipitated NiS mineral particles in BSA. These samples were all spotted on gold-coated silicon wafers and analyzed by ToF-SIMS as described above in both positive and negative mode.

**Stable isotope labeling experiments**. Stable isotope labeling was performed by adding $^{13}C$ labeled glucose and $^{34}S$-labeled sulfate to sediments in which cable bacteria were grown. For the $^{13}C$-glucose experiment, 5 mg of glucose was added to each core (42 mm internal diameter) and mixed into the top ~3 cm of the core immediately after it was repacked. These cores were incubated submerged in an aerated aquarium. For the $^{34}S$-sulfate labeling experiment, ASW was prepared to a salinity of 20 (to match in situ salinity) and 20 mM 90 atom % $^{34}S$-sulfate. Before repacking, 25 ml sediment was centrifuged at 110 g for 5 min (supernatant discarded), re-suspended in 25 ml $^{34}S$-labeled ASW then repacked into 1 cm diameter cut off syringes. Once the sediment had consolidated, the syringes were topped up with $^{34}S$-labeled seawater (~5 cm water column) and a bubbler line was added to each core. Cores were regularly topped up with ultra-pure water to compensate for evaporation. For both labeling experiments a control with unlabeled substrates was treated identically. For $^{13}C$, filaments were collected for Raman imaging at 14 and 21 days after labeling. In the case of $^{34}S$, filaments were collected after 20 and 50 days for Raman imaging.

**The role of the Ni/S group in electron conduction**. Two types of experiments were performed to determine the role of the Ni/S group in the conduction of the fiber sheaths. First, the effect of oxidation or reduction on the Raman signal of the two lower bands at 367 and 492 cm$^{-1}$ was determined. For the Raman measurements, extracted fiber sheaths were pretreated for 10 min in 20 μl droplets of the reductant (10 mM K$_4$Fe$^{II}$(CN)$_6$) or oxidant (10 mM K$_3$Fe$^{III}$(CN)$_6$) and enclosed in the same solute between a glass microscope slide and cover slip sealed with nail polish. At least 15 spectra from five individual fiber sheaths were recorded with a 532 nm green laser Raman microscope from Renishaw as described above. Subsequently, the effect of oxidation or reduction on the conductance of fiber sheaths was evaluated. To this end, individual fiber sheaths were stretched out on a microscope cover slip and a small spot of conductive water-based carbon paste (EM-Tec) was added as an electrode on both ends of the filament. After the carbon paste electrodes had dried, they were connected with conductive copper tape and additional carbon paste to a Palmsens 4 potentiostat and sealed with nail polish to make them water resistant. The effect of reduction or oxidation on filament conductance was determined by adding 2 μl of 10 mM K$_4$Fe$^{II}$(CN)$_6$ or 10 mM K$_3$Fe$^{III}$(CN)$_6$ respectively to the fiber sheath area between the electrodes. As control we added a 2 μl droplet of MilliQ. The electrical current was measured directly before and after drop addition. Adding a second (i.e., fresh) 2 μl droplet of oxidant or reductant typically did not result in a further change in conduction. We performed repeated oxidation, reduction and MilliQ cycles on single filaments. Occasionally unexpected, sharp drops in conduction were seen between treatments, suggesting the fiber sheath had been damaged during exchange of solutes. These suspect data were excluded from the analysis. Conductance measurements were performed at room temperature and current–voltage (IV) curves were recorded (bias −0.1 to 0.1 V, scan rate 0.01 Vs$^{-1}$). The length (Δx) of the filament between the electrodes was measured with a Dino-lite microscope. Normalized current was calculated as $I_{norm} = I × (\Delta x/\Delta x_{ref})$ with $\Delta V = 0.1$ V and $\Delta x_{ref} = 0.3$ mm (see[7]).

In a second set of experiments, Ni was removed selectively by extraction in 50 mM EDTA and effects on Raman signals and conduction were determined. For the Raman measurements, SDS only extracted fiber sheaths were treated for 10 or 40 min in 20 μl droplets of 1 or 50 mM EDTA and deposited after 5 MilliQ washes on a glass microscope slide. At least 15 spectra from five individual fiber sheaths were recorded with a 532 nm green laser as described above. Additionally, the conductance of ten individual extracted fiber sheaths that were either treated with the standard 10 min 1 mM EDTA or 10 min 50 mM EDTA were compared. Conductance was measured in terms of normalized current as described above. A 10 min extraction time was used for both treatments as this resulted in similar handling times. Fresh EDTA solutions were prepared daily from 0.5 M EDTA stock solution (pH 8) and filaments were treated in batches of five filaments to reduce the exposure to oxygen. Morphology of the fiber sheaths treated with 1 mM or 50 mM EDTA was studied by AFM on a Multimode 8 microscope (Bruker, Santa Clara CA, USA) in the peak-force quantum nano mechanical mode[5].

**Reporting summary**. Further information on research design is available in the Nature Research Reporting Summary linked to this article.

## Data availability
Detailed ToF-SIMS and SDM results and discussion are available in Supplementary Notes 1–2. Other relevant data will be made available by the authors upon request. Source data are provided with this paper.

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

## Acknowledgements

The authors thank Marlies Neiemeisland for assistance with Raman microscopy, Michiel Kienhuis for assistance with NanoSIMS analysis, Peter Hildebrandt and Diego Millo for helping with the interpretation of the Raman spectra, IONTOF for the Orbitrap Hybrid-SIMS analysis, and Rene Fabregas for helping with finite-element numerical modeling for SDM. H.T.S.B. and F.J.R.M. were financially supported by the Netherlands Organization for Scientific Research (VICI grant 016.VICI.170.072). Research Foundation Flanders supported F.J.R.M., J.V.M., and R.T.E. through FWO grant G031416N, and F.J.R.M. and J.S.G. through FWO grant G038819N. N.M.J.G. is the recipient of a Ph.D. scholarship for teachers from NWO in the Netherlands (grant 023.005.049). The NanoSIMS facility at Utrecht University was financed through a large infrastructure grant by the Netherlands Organization for Scientific Research (NWO, grant no. 175.010.2009.011) and through a Research Infrastructure Fund by the Utrecht University Board. A.G.S. is supported by the Special Research Fund (BOF) of Ghent University (BOF14/IOP/003, BAS094-18, 01IO3618) and FWO (G043219). The ToF-SIMS was funded by FWO Hercules grant (ZW/13/07) to J.V.M. and A.F. H.L., R.M.S., and G.G. were funded by the European Union H2020 Framework Programme (MSCA-ITN-2016) under grant agreement n 721874.EU, the Spanish Agencia Estatal de Investigación and EU FEDER under grant agreements TEC2016-79156-P and TEC2015-72751-EXP, the Generalitat de Catalunya through 2017-SGR1079 grant and CERCA Program. G.G. was recipient of an ICREA Academia Award, and H.L. of a FPI fellowship (BES-2015-074799) from the Agencia Estatal de Investigación/Fondo Social Europeo. L.F. received funding from the European

Research Council (grant agreement No. 819417) under the European Union's Horizon 2020 research and innovation programme.

## Author contributions

H.T.S.B. and F.J.R.M. designed the study and were involved in all experiments, measurements, and data analysis. S.H.M. performed filament cultivation and fiber sheath extraction. S.H.M., H.T.S.B., N.V.G., and H.R. were involved in the Ni group redox and high EDTA experiments. P.L.M.C., K.K., M.M., and B.W. performed the stable isotope labeling experiments. L.P. and N.M.J.G. performed the NanoSIMS analysis. H.T.S.B., R.T.E., V.S., A.F., and J.V.M. carried out the ToF-SIMS analysis. H.T.S.B., P.L.M.C., K.K., B.W., D.K., J.T.B., and A.G.S. performed the Raman analysis. N.C., P.K., D.W., and S.B. carried out the HAADF-STEM and STEM-EDX analysis. K.K.S., L.P.N., F.C., and T.H. were involved in AFM-IR analysis. D.B., A.G., and S.H.M. carried out the LEXRF measurements. H.T.S.B. and F.J.R.M. developed the chemical model of the fiber sheath with additional input from J.S.G., N.V.G., and H.R. H.L. and R.M.S. carried out the SDM experiments and analyzed the results. L.F. and G.G. developed the SDM methods and supervised the interpretation of the SDM results. H.T.S.B. and F.J.R.M. wrote the paper with contributions provided by all co-authors.

## Competing interests

The authors declare no competing interests.
