## [Peer Review File · Nature Communications]

REVIEWER COMMENTS

Reviewer #1 (Remarks to the Author):

Outstanding high impact manuscript that should be published without delay. The potential role of Ni in ultra-long distance electron transfer is a huge discovery – made more so by the high copper wire-like electrical conductivity. One small suggestion – it would be great if the authors could comment on the potential role of the cytochromes in the fibers mentioned in line 203. Also, I would fully expect conductivity to depend on the redox/charge state of the fibers. If a classic redox conductor – then I would expect low conductivity when either fully reduced or fully oxidized, with highest conductivity at 50/50 mixed state. Can it be that your reduced state is not totally reduced due to exposure to air? Do the measurements change over time for reduced filaments?

Reviewer #2 (Remarks to the Author):

Boschker et al. used mainly spectroscopic techniques to identify chemical signatures for fiber sheaths of cable bacteria and propose a “core-shell” model to explain their conductivity. The model assumes that the fibers have a protein core surrounded by an insulating layer that they speculate could be of exopolysaccharide. The protein core is said to be rich in aromatics but their potential for conductivity is not investigated nor discussed. The IR spectra suggests that the core contains disordered proteins rather than the helical peptides expected if these fibers were assemblies of pilin peptides, as previously proposed by the team. The detection of Ni and S in the fibers is interpreted as indicating that the protein core uses thiol groups to bind Ni and that the metal is “essential” for fiber conductivity. I appreciated the detailed spectroscopic study and advanced methodologies used by the team to probe the chemistry of these structures. I am concerned, however, that the model relies quite heavily on speculation and overstated results. Alternative explanations are missed all together, weakening the validity of the proposed model. Further lessening my enthusiasm is the fact that the model is not integrated with past models and experimental evidence presented by the team. First, they proposed conduction via cytochromes, then boldly proposed that the fibers were bundles of pili assembled as periplasmic fibers. Now it is an unknown protein core doped by Ni. I am ok with proposing working models but the experimental foundation needs to be more robust.

COMMENTS:

- Line 141: “This suggests that the fiber sheath is made of a protein layer on top of a basal polysaccharide-rich layer.”
 - o How could an EPS-rich layer be secreted into the periplasmic space? This is quite speculative and not discussed. For example, could it be that the proteins in the fiber’s core are glycosylated?
- Line 143: “Together, the HAADF-STEM, AFM-IR and ToF-SIMS data show that (i) the conductive fibers are positioned in a regular, parallel pattern on the outside of the fiber sheath, (ii) that the fibers consist of protein that is rich in aromatic amino acids, 145 and (iii) that the fibers rest upon a basal sheath rich in polysaccharide”. Missing in this summary is the finding that the fibers are not pili, challenging a model previously proposed by the team.
- Line 173: Raman spectra of purified fiber sheaths “showed no cytochrome signal, thus confirming that the conduction mechanism does not involve cytochromes”.
 - o But the cytochromes were proposed to be the primary pathway for in vivo conductivity along the filaments. The authors need to reconcile the in vivo and in vitro results.
 - o There is some Fe in these fibers (LEXRF maps), where is it coming from and what role does it play?
- Line 198: “Metals commonly found in metalloproteins were present in intact filaments, but concentrations were low and close to detection limits: Fe (0.033-0.047 Atm%), Ni (0.009 Atm%) and Cu (0.006-0.009 Atm%). After fiber sheath extraction, Ni (0.016-0.037 Atm%) was selectively enriched by a factor of 2-4 compared to intact filaments.”
 - o The low levels of Ni in the intact filaments argue against an essential role in in vivo conductivity.
 - o How does the Ni enrichment compare to the fiber enrichment? (i.e., was Ni enriched proportionally to the fiber increases. This would be expected if the fibers have defined surface

motifs for metal binding. Otherwise Ni could have been enriched artifactually during purification.)

- Figure 3: Elemental analyses is used to conclude that the fibers are Ni-rich.
 - o The authors state that the intact cables and fiber sheaths show “a detectable Ni signal and lower Fe and Cu levels in the fiber sheath”. But Fig. 3E shows higher levels for Cu and the enrichment of this metal in the fiber sheaths. Why was Cu metallization of the protein core not considered in this study?
 - o Could the fiber extraction protocol affect the metal content?
- Line 254: The EDTA experiments are critical to demonstrate a role for Ni in conductivity. I note some weaknesses in the experimental approach that challenge this interpretation.
 - o EDTA is not a specific metal chelator and will chelate other metal cations. Yet, the authors conclude that “Ni was selectively removed”.
 - o Did they check the elemental composition of the EDTA extract to see what metals were removed? The Raman signal for Ni only dropped to 45% but conduction dropped to 62%.
 - o Fig. 4C is missing a control treatment with EDTA only.
- In the Supplementary file (line 337) they state that “Fiber sheaths display an exceptional chemical resistance, as they remain [retain] their integrity and conductivity after SDS and EDTA treatments, and this resistance could be aided by protein disulfide bridges”.
 - o How do they know that the fiber sheath structure was not affected during purification?
 - o If disulfide bonds are so important for structure and conductivity, the chemical reduction of the fiber sheath iron cyanide would have affected these fiber properties. Yet, the authors indicate that, “reduced fiber sheaths showed a 2.1 ± 0.5 (N = 11) higher conductance than oxidized fiber sheaths” (line 248, main text).
- Discussion: Overstatements abound. I note some examples:
 - o Line 302: “Our results demonstrate that conduction in cable bacteria occurs through proteins with Ni-dependent cofactors”.
 - o Line 332: “At this moment, the exact mechanism of conduction remains unclear, but our results demonstrate that the novel Ni-cofactor is an essential component”.
 - o Line 334: “... we also detected substantial signals from aromatic amino acids in the fiber proteins, one possibility is that conduction is based on electron transfer between S-ligated Ni-groups assisted by bridging aromatic groups in nearby aromatic amino acids”.
 - ♣ What does “substantial” mean in this context? (Note that there are many aromatic-rich proteins that are not conductive.)
 - ♣ A model of hybrid conduction involving S-ligated Ni and aromatic residues requires knowledge of the distribution of metal and aromatic clusters. The LEXRF maps show metal clusters but the organization of aromatics is not known. Furthermore, the aromatic clusters would have to bridge the large distance between the metal clusters.
 - ♣ How does Cu fit in this model? (Note the Cu clusters).
 - o Line 339: “... promising gateway for new technology, and creates the prospect of bio-electronic devices with new functionality that integrate proteins as new class of electronic materials.” Big overstatement at the end of the discussion when they do not even know whether this is an organized assembly and what proteins make it. It also surprises considering that technologies based on the *Geobacter pili* have been realized already.

Reviewer #3 (Remarks to the Author):

Boschker and coauthors describe a set of analytical experiments to determine the chemical features of cable bacteria sheaths that support long-range electronic conductivity. In my opinion, their conclusions are well supported and this manuscript represents a significant advance to understanding the structural and chemical underpinnings of the most conductive and longest microbial structures characterized to date. My comments are relatively minor and I imagine they could be readily addressed by the authors.

1. First, a general comment about softening the language used throughout the manuscript. The area of research focusing on understanding the chemical and structural features that allow for long-range electronic conductivity in microbial structures has undergone an upheaval in the past several years, with new structures and new understanding calling into question many long-held beliefs about what is or is not possible and providing new insights into how anaerobes respire. The

understanding of why these cable bacteria sheaths are so electrically conductive has been and will continue to be a challenge. Even with this manuscript, which is a significant advancement towards understanding the working principles of these sheaths, the field still lacks a molecular and genetic basis of these structures. Lacking these more conclusive pieces of evidence, I highly recommend softening the language in the paper to present results as "suggesting" or "consistent with" or "providing further support for" a hypothesized model. For example, the usage of "indicate/s/d/ing" in lines 110, 124, 180, 186, 257, 365, 394 and "confirm/s/ed/ing" in lines 107, 173, 205, 210, 255, 259 should be softened. Many of the experiments in this work are indirect and subject to artifacts. To keep the presentation of results objective, I suggest a rewording of the implications of these outcomes. The data are consistent with a model, but do not prove it conclusively, and should be presented as such.

2. Line 119-126 ignores recent results suggesting a role other than conductivity for Pila aromatic species. Aside from the problematic lack of an experimentally determined atomistic structure that supports a chemical and structural basis for electronic conductivity in *G. sulfurreducens* Pila, Yalcin et al Nat Chem Biol 16, 1136-1142, 2020 show that mutations to aromatic amino acids affect cytochrome secretion which in turn affects the proportion of more or less conductive cytochrome wires. So while the speculation on lines 333-336 that aromatic amino acids may bridge metal-sulfide cofactor sites is not off base, the rationale that aromatic amino acid content alone can be responsible for conductivity as suggested in 119-126 needs to be better contextualized.

3. Related to the previous point, the appearance of aromatic signal in IR and ToF-SIMS does not necessarily mean aromatic amino acids are more abundant than in a typical protein. Comparisons to control protein signals by the same techniques, either by additional experiments or comparison to literature data would help contextualize the assessment that the signal constitutes an "aromatic rich" protein.

4. The core-shell fiber model seems reasonably supported by evidence, but it is unclear how it works in a biological context. If the conductive elements are shielded from the surrounding environment, how do the *Desulfobulbaceae* dump electrons into the structures in the anoxic zone, how does molecular oxygen access the reducing potential in the oxic zone, and how do other anaerobes access the oxidative potential of the cables in the anoxic zone, as has been shown elsewhere? The selective insulating properties of the shield layer should be clarified.

5. Figure 4D is confusingly presented. Do the lines represent before and after EDTA treatment currents of individual sheaths? How are the sheaths arranged left to right, by decreasing current reading? In my opinion the information would be more effectively presented as a comparison of the means (+/-standard error) of the "before" samples and "after" samples. Alternatively, it could be presented as the mean change in current in the "before" samples and "after" samples. Either way, the current presentation makes it difficult to interpret.

6. The statistical methods used are unfamiliar to me. Could the authors explain why the Wilcoxon test was used instead of the more commonly used Student's t-test?

7. Lastly, is there a genetic basis for thinking that Ni-S metalloenzymes are made by *Desulfobulbaceae*? If the authors could establish that finding Ni-S containing proteins in these organisms is expected from genome analysis, that would bolster the molecular model they are building with this work.

Otherwise, the manuscript is excellent and represents a substantial advance in the understanding of the structures supporting long-range electronic conductivity in these exceptional bacteria.

DETAILED RESPONSE TO THE REVIEWER COMMENTS NCOMMS-20-41197A (the response in marked in blue font , and line numbers refer to the annotated version of the revised manuscript)

Reviewer #1 (Remarks to the Author):

Outstanding high impact manuscript that should be published without delay. The potential role of Ni in ultra-long distance electron transfer is a huge discovery – made more so by the high copper wire-like electrical conductivity.

We thank the reviewer for the suggestions and general support for the manuscript.

One small suggestion – it would be great if the authors could comment on the potential role of the cytochromes in the fibers mentioned in line 203.

The selective loss of Fe during extraction of the fiber sheaths, is in agreement with the loss of cytochrome bands in the Raman spectra. We have now clarified this by adding “which is consistent with the loss of cytochrome bands in the Raman spectra after extraction (Fig. 2A)” (line 212).

We also now additionally discuss the potential role of periplasmic cytochromes in the transfer of electrons to and from the fibers (line 314, see also our response to comment 4 of Reviewer 3):

“However, the insulating outer layer also provokes questions on how electrons generated Cable bacteria must contain a mechanism for this electron transport to and from the conductive fiber core, which may involve periplasmic cytochromes¹² that are present in intact bacteria, but removed during fiber sheath extraction (Fig. 2A and 7).”

Also, I would fully expect conductivity to depend on the redox/charge state of the fibers. If a classic redox conductor – then I would expect low conductivity when either fully reduced or fully oxidized, with highest conductivity at 50/50 mixed state. Can it be that your reduced state is not totally reduced due to exposure to air? Do the measurements change over time for reduced filaments?

This is a good point. For a true redox conductor, one would expect a large effect of chemical oxidation and reduction on conduction, and maximum conduction at intermediate redox states. Yet, we observe that the effect of chemical oxidation and reduction on conduction is relatively small, reduced fiber sheaths are about 2 time more conductive. Maybe the Ni group doesn't operate as a traditional redox group during electron transport although we show that it can be oxidized and reduced reversibly. This aspect should be clarified in further studies as we also note in the discussion (line 366).

There is indeed a slow decay of the conduction with time (~ hours) in reduced fiber sheaths as they are apparently sensitive to oxygen exposure (but much less sensitive than the intact cable bacterium filaments). Note that measurements were always done right after treatment, to avoid impact of this slow decay.

We also explicitly tested the refreshing of the reduction and oxidation agents (see Methods), which did not change electron conduction further. But we are indeed not certain that the Ni group was completely reduced as measurements were done in air. We have therefore added now “Note that the Ni group is possibly not completely reduced, as measurements were done in air, and so the effect of oxidation state on conductance may actually be larger.” (line 272).

Reviewer #2 (Remarks to the Author):

We thank the reviewer for the in-depth comments that certainly helped to improve the manuscript.

Boschker et al. used mainly spectroscopic techniques to identify chemical signatures for fiber sheaths of cable bacteria and propose a “core-shell” model to explain their conductivity.

The model assumes that the fibers have a protein core surrounded by an insulating layer that they speculate could be of exopolysaccharide.

This is a misinterpretation of the model that we propose. We do not state that the insulating layer surrounding the conductive core is made of polysaccharide or is polysaccharide-rich. In fact, all our data suggest (ToF-SIMS and SDM analysis) that the outer insulating shell is made of protein. The core-shell model of the fibers is described as an outer protein (not polysaccharide) shell layer that is non-conductive and an inner core of Ni and S rich protein that is conductive. The whole fiber structure is then embedded in polysaccharide-rich basal layer that is rich in polysaccharide.

We clearly state in the discussion of the model that the polysaccharide layer is most likely a peptidoglycan layer (and not an exopolysaccharide) that is commonly found with the periplasmic space of bacteria with a Gram-negative cell envelope such as the cable bacteria. And that the conductive fibers are sitting on top or are partially embedded onto this peptidoglycan layer.

The protein core is said to be rich in aromatics but their potential for conductivity is not investigated nor discussed. In theory, the role of aromatics could be investigated by genetic engineering and amino acid swapping.

It is currently not possible to investigate the role of aromatics by genetic engineering and amino acid swapping in cable bacteria. Cable bacteria are complex multicellular bacteria that so far can only be grown in natural sediments, and a system to genetically engineer cable bacteria (such as available for *Geobacter*) does not yet exist.

In the discussion of the original manuscript, we speculated briefly that the aromatic amino acids as detected in the fiber sheaths may be involved in conduction in combination with the Ni-group. We have toned this conclusion in response to this reviewer’s comments below and the comment 3 from Reviewer 3. So with the current available dataset and methods, we cannot determine what the role is of the aromatics.

The IR spectra suggests that the core contains disordered proteins rather than the helical peptides expected if these fibers were assemblies of pilin peptides, as previously proposed by the team.

The detection of Ni and S in the fibers is interpreted as indicating that the protein core uses thiol groups to bind Ni and that the metal is “essential” for fiber conductivity. I appreciated the detailed spectroscopic study and advanced methodologies used by the team to probe the chemistry of these structures.

I am concerned, however, that **the model relies quite heavily on speculation and overstated results. Alternative explanations are missed all together, weakening the validity of the proposed model.**

As shown below (response to detailed comments), we believe that this assessment is largely based on a misunderstanding of the fiber model that we propose. In this revised version, we have now better explained this fiber model in order to remove such misunderstandings.

Further lessening my enthusiasm is the fact that the model is not integrated with past models and experimental evidence presented by the team. First, they proposed conduction via cytochromes, then boldly proposed that the fibers were bundles of pili assembled as periplasmic fibers. Now it is an unknown protein core doped by Ni. I am ok with proposing working models but the experimental foundation needs to be more robust.

The fiber model proposed is not in conflict with past evidence presented by our research team. Foremost, we never proposed a conduction mechanism based on cytochromes, on the contrary; Bjerg et al 2018 revealed cytochromes in living cable bacteria, but did not claim they were involved in conduction; Meysman et al 2019 clearly demonstrated that cytochromes were **not** involved in conduction. Secondly, conduction via PilA structures was brought forward as an hypothetical model (derived from genome and proteome interpretation) that needed experimental confirmation (Kjeldsen et al 2019).

Yet, we do agree with the referee that past work on and proposed mechanisms for conduction were not well discussed and put into context. We have now improved this by adding additional discussion (see below for responses to specific comments).

COMMENTS:

- Line 141: “This suggests that the fiber sheath is made of a protein layer on top of a basal polysaccharide-rich layer.”

o How could an EPS-rich layer be secreted into the periplasmic space? This is quite speculative and not discussed. For example, could it be that the proteins in the fiber’s core are glycosylated?

We do not propose that an Extracellular Polymeric Substance (EPS) layer surrounds the fibers. Instead, we propose that the fibers are embedded in basal polysaccharide-rich layer, that is most likely the peptidoglycan layer as commonly found in bacteria with a Gram-negative cell envelope like cable bacteria.

To clarify this, we now discuss the peptidoglycan layer already with the ToF-SIMS data (line 150):

“In bacteria with a Gram-negative cell envelope such as cable bacteria, the most likely source of the polysaccharide layer is the periplasmic peptidoglycan layer (see supplementary text for further discussion). Cable bacteria genomes¹² indeed contain genes encoding for penicillin-binding proteins that perform the final steps in peptidoglycan biosynthesis¹⁸.”

And repeat this argument at the end of the paragraph where the core-shell model is introduced (line 301):

“As argued before, this polysaccharide-rich layer is most likely the peptidoglycan layer as commonly found in bacteria with a Gram-negative cell envelope such as cable bacteria¹⁸.”

In the abstract, we also made clear that the shell layer is made of protein (line 53): “the periplasmic wires consist of a conductive protein core surrounded by an insulating **protein** shell layer”

- **Line 143: “Together, the HAADF-STEM, AFM-IR and ToF-SIMS data show that (i) the conductive fibers are positioned in a regular, parallel pattern on the outside of the fiber sheath, (ii) that the fibers consist of protein that is rich in aromatic amino acids, 145 and (iii) that the fibers rest upon a**

basal sheath rich in polysaccharide". Missing in this summary is the finding that the fibers are not pili, challenging a model previously proposed by the team.

The absence of pili was based on our interpretation of the Amide band position. Yet, the Amide band should be interpreted with caution. The fibers only contribute 25% of the fiber sheath cross-section (detailed calculations are newly added to the supplementary text) and so the Amide I band in the AFM-IR spectra will therefore originate from a mixture of the fiber proteins and other proteins within the fiber sheath. This makes it difficult to draw any strong conclusions on the secondary structure of the proteins in the fibers, and therefore, we do not include "that the fibers are not pili" as a hard conclusion in the summary.

The discussion on the Amide I band shape now reads as follows (line 126):

"The position of the Amide I peak at 1643 cm^{-1} suggests however that the protein secondary structure is mainly disordered, and not of the α -helix type¹⁰, which hence speaks against an abundance of α -helix rich pili. We estimate that the fibers themselves represent 25% of the total mass of the fiber sheath (see supplementary text), and hence the Amide I peak could not only originate from fiber proteins, but also from other proteins within the fiber sheath. Therefore, the potential absence of pilin protein in the fibers requires further confirmation."

- Line 173: Raman spectra of purified fiber sheaths "showed no cytochrome signal, thus confirming that the conduction mechanism does not involve cytochromes".

o But the cytochromes were proposed to be the primary pathway for in vivo conductivity along the filaments. The authors need to reconcile the in vivo and in vitro results.

We disagree that cytochromes were ever proposed to be the primary conduction pathway. In Meysman et al 2019, we already showed that the extracted fiber sheaths show no cytochrome Raman signal anymore, thus excluding a role for cytochromes in the conductive fibers. In the present manuscript, we confirm that the fiber sheaths contain no cytochrome Raman signal anymore (Fig 2A, line 173 in the submitted version is line 181 in the revised manuscript).

The reviewer may refer to Bjerg et al 2018, where we used Raman microscopy to study cytochrome signals and oxidation state along active cable bacteria. In that paper, we discussed 3 possible roles for the detected cytochromes. The first role discussed is that cytochromes could theoretically be part of the conductive structure, which would then suggest a mechanism of electron hopping via heme groups along the redox gradient. However, we also immediately stated that the electron-hopping frequency and the amount of heme groups required to explain the observed rates of electron transport would be unprecedented, making this role very unlikely. So we concluded that the cytochromes likely had another role, which is the up- and downloading of electrons to the fibers, as now added to the revised manuscript (line 313 and further, see specific response to the comments of Reviewer 1 and comment 4 of Reviewer 3).

o There is some Fe in these fibers (LEXRF maps), where is it coming from and what role does it play?

The LEXRF maps and counts show that Fe is removed during the extraction of the fiber sheaths and that remaining Fe counts are low. This is in agreement with the STEM-EDX data that also show a loss of Fe. And it was finally confirmed by the ToF-SIMS data, where intact cable bacteria show high Fe counts in the periplasmic space, whereas the counts in the extracted fiber sheaths are low especially

compared to the Ni counts. We don't know if the remaining low amounts of Fe has any role in the fiber sheaths in terms of electron conduction.

- Line 198: "Metals commonly found in metalloproteins were present in intact filaments, but concentrations were low and close to detection limits: Fe (0.033-0.047 Atm%), Ni (0.009 Atm%) and Cu (0.006-0.009 Atm%). After fiber sheath extraction, Ni (0.016-0.037 Atm%) was selectively enriched by a factor of 2-4 compared to intact filaments."

o The low levels of Ni in the intact filaments argue against an essential role in in vivo conductivity.

The Ni value in the intact filaments is the bulk value in the biomass, which is indeed low. However, our results suggest that Ni is concentrated in the conductive core of the fibers, which only accounts for a small fraction of the total biomass. Based on AMF data in Cornelissen et al 2018, we estimate that the fiber sheath explains 25% of the total biomass. And we added new calculations (see supplementary text), which reveal that the conductive core of the fibers is actually only 7% of the total fiber sheath at the cell area. To conclude, the actual Ni concentrations in the fiber cores are therefore 13 times higher than the bulk value of the fiber sheath and about 50 times higher than in the intact cable bacteria filaments.

The distance between the metal centers is the main parameter determining electron transfer rates between metal centers in proteins. To further put the Ni content into context, we therefore estimated the distance between Ni atoms assuming that all Ni is located in the conductive core of the fibers and that Ni is homogeneously distributed within this core. We arrive at distance between Ni atoms of 1.4 - 1.9 nm, which is in the range commonly observed for electron transfer in metalloproteins. However, if one assumes that electron hopping between the Ni-groups is the main conduction mechanism, the estimated electron tunneling times for the Ni group (50-90 nsec) must be still rather fast suggesting that other conductive structures must be present.

We have now added a detailed description of the calculations of the interatomic Ni distances in the supplementary text (line 491), and discuss them in comparison to what is known from *Geobacter* nanowires in the revised and extended discussion (line 407):

"The exact mechanism of electron transport in the periplasmic fibers of cable bacteria remains unclear, but our results indicate that the novel Ni-cofactor is an essential component. Electron transport in metalloproteins critically depends on the distance between metal centers⁴², and assuming an homogeneous distribution of Ni atoms throughout the conductive fiber core (see supplementary text), the distance between Ni atoms is 1.4-1.9 nm, which leads to a characteristic electron tunneling time of 50-90 nsec. This interatomic Ni distance is in the range commonly found for metal centers in metalloproteins, but the electron tunneling times are on the fast end of what has been observed for single electron transfers⁴². In comparison, the interatomic distance between Fe atoms in OmcZ cytochrome based nanowires from *Geobacter* is somewhat smaller (on average 1.2 nm) and tunneling times are only approximately 5 nsec (both calculated from²²). It has been argued that the high conductivity of OmcZ nanowires is primarily due to the enhanced pi/pi-stacking of closely arranged heme groups. The fast electron tunneling times of 50-90 nsec suggests that the conductive fiber core must possess a supramolecular organization that supports highly efficient electron transport between Ni centers. This could involve particular configurations of aromatic amino acids, or maybe an unknown prosthetic group involved in Ni-ligation. Future studies should further elucidate the genetic and molecular basis of the highly efficient long range electron transport in cable bacteria."

o How does the Ni enrichment compare to the fiber enrichment? (i.e., was Ni enriched proportionally to the fiber increases. This would be expected if the fibers have defined surface motifs for metal binding. Otherwise Ni could have been enriched artifactually during purification.)

Good suggestion. Based on AFM mapping (as done in Cornelissen et al 2019), we can crudely estimate the loss of biomass upon extraction. We have now added to the text (line 210):

“Based on AFM mappings⁵, fiber sheaths contain ~4 times less biomass compared to intact filaments, thus explaining selective Ni enrichment.”

We are very much aware that impurities could be an issue given the relatively low metal concentrations and the small samples. However, it is unlikely that the Ni was added during the fiber sheath extraction because:

- i) the Ni content of the intact bacteria and the fiber sheaths were similar (LEXRF data Fig 3), so no additional Ni was added. It should be noticed that LEXRF is based on X-rays that will penetrate the thin samples completely. LEXRF data are therefore integrated over the whole sample.
- ii) Ni is mainly found in a sharp subsurface peak in the fiber sheaths (ToF-SIMS data Fig 1), whereas impurities would most likely be mainly found at the surface (i.e. like Cu).
- iii) We also found a similar subsurface Ni (and S) peak in the periplasmic space in the ToF-SIMS profiles of the intact cable bacteria (supplementary Fig. 11) showing that the Ni was already present before the fiber sheath extraction. We have also transferred the discussion of the ToF-SIMS results from the intact cable bacteria filaments from the supplementary text to the main text to increase their visibility to the reader (line 232).

• Figure 3: Elemental analyses is used to conclude that the fibers are Ni-rich.

o The authors state that the intact cables and fiber sheaths show “a detectable Ni signal and lower Fe and Cu levels in the fiber sheath”. But Fig. 3E shows higher levels for Cu and the enrichment of this metal in the fiber sheaths. Why was Cu metallization of the protein core not considered in this study?

The caption text “a detectable Ni signal and lower Fe and Cu levels in the fiber sheath” belongs to the STEM-EDX results in Fig 3 A and B, and not the LEXRF results presented in Fig. 3E. We indeed found high and variable Cu content in the LEXRF analysis but argue that this is likely an impurity as i) Cu is substantially lower than Ni and Fe in the STEM-EDX, ToF-SIMS and NanoSIMS data and that ii) Cu shows a surface peak in the ToF-SIMS data indicating that suggests an impurity. We currently do not know what the source of this Cu impurity is and why it is mainly seen in the LEXRF analysis, though not in the other metal data from STEM-EDX, ToF-SIMS (although ToF-SIMS shows that Cu is mainly found in a surface peak indicating a possible impurity as was discussed in the supplementary text) and NanoSIMS analysis.

o Could the fiber extraction protocol affect the metal content?

It certainly was a concern that the metal content in the fibers could be affected by the fiber sheath extraction protocol. This is especially true for the second 1 mM EDTA extraction step in the standard SDS+EDTA protocol as EDTA is a ligand that is known to bind metal ions. However, Raman signals from the two lower bands were the same in the SDS only and the standard SDS-EDTA extractions suggesting that the EDTA step had no effect on the Ni-group (Fig 4C). In addition, the Ni content in

the intact cable bacteria filaments is similar to the extracted fiber sheaths as shown by the LEXRF analysis (Fig. 3E), suggesting that no Ni was removed or added. Only the high 50 mM EDTA extractions reduced the Raman signals from the two lower bands significantly (Fig 4C).

We added the following text (line 281):

“EDTA is a known mobilizing agent for metals in biostructures. Raman signals from the SDS only treatment and the standard SDS+EDTA extraction were similar (Fig. 4C) suggesting that an EDTA treatment at low (1 mM) concentrations did not significantly affect the Ni/S group.”

- Line 254: The EDTA experiments are critical to demonstrate a role for Ni in conductivity. I note some weaknesses in the experimental approach that challenge this interpretation.

o EDTA is not a specific metal chelator and will chelate other metal cations. Yet, the authors conclude that "Ni was selectively removed".

Alternatively, the high (50 mM) EDTA treatment could also remove other metals (e.g. Ca and Mg) from the fiber structures, which could degenerate secondary and tertiary protein structure and also cause a decrease in conductivity.

We agree with the referee that this was not formulated correctly. It is clearly stated though that removal of the Ni is based on the Raman data that are specific for the Ni-group. However, even though the fiber structure did not change, we can indeed not exclude that other metals were not affected by the high EDTA treatment.

We have reworded this to (line 278):

“Additional experiments, in which Ni was partially removed from the fiber sheath through extraction with high EDTA concentrations, provided further support that the Ni/S-group plays a crucial role in electron transport. EDTA is a known mobilizing agent for metals in biostructures. Raman signals from the SDS only treatment and the standard SDS+EDTA extraction were similar (Fig. 4C) suggesting that an EDTA treatment at low (1 mM) concentrations did not significantly affect the Ni/S group. However, extraction with 50 mM EDTA caused a decreased the Raman signal by 45% suggesting that Ni was selectively removed (Fig. 4C). Fiber structure remained intact (Supplementary Figure 6), but we can however not fully exclude that other metals (e.g. Ca and Mg) were also partially removed from the fiber structures by the high (50 mM) EDTA treatment, which may have affected secondary and tertiary protein structures to some extent. At the same time, the 50 mM EDTA treatment reduced the conduction on average by 62% (Fig. 4D), which hence could suggest that the Ni/S-group plays a key role in maintaining high rates of long-distance electron transport in cable bacteria.”

o Did they check the elemental composition of the EDTA extract to see what metals were removed? The Raman signal for Ni only dropped to 45% but conduction dropped to 62%.

We did not check the metal concentrations in the EDTA extract. This is not analytically possible, as volumes are small (ca. 20 microliters) and the concentration will be very low, as there are only a couple of cable bacteria filaments in these extractions.

o Fig. 4C is missing a control treatment with EDTA only.

This control treatment is not needed as we do not present any data where intact cable bacteria are extracted with EDTA only. The first SDS extraction step is the main step in the standard extraction

protocol as it removes the outer and cell membranes and most of the cytoplasm, thus leaving what we call the fiber sheath. The 1 mM EDTA step was added to remove any SDS that remained absorbed onto the sample after washing steps.

- In the Supplementary file (line 337) they state that “Fiber sheaths display an exceptional chemical resistance, as they remain [retain] their integrity and conductivity after SDS and EDTA treatments, and this resistance could be aided by protein disulfide bridges”.

o How do they know that the fiber sheath structure was not affected during purification?

It is difficult to fully exclude any effect of the fiber sheath structure during the standard SDS-EDTA extraction. However, we have shown that both the intact cable bacteria and the extracted fiber sheaths are highly conductive (Meysman et al 2019), suggesting that the conductive structures remain intact. In fact, the conduction appears to be somewhat higher on average in the fiber sheaths (Meysman et al 2019), but this may be due to the much slower decay on conduction in the fiber sheaths with time (as also shown by Meysman et al 2019). In addition, the fiber network remains intact as shown by Cornelissen et al 2018 and in the STEM images in the current manuscript (Fig 1A and supplementary movie 1).

Clarification made in the main text (line 246):

“Fiber sheaths display an exceptional chemical resistance, as the fiber extraction protocol uses relatively strong chemical agents (SDS and EDTA), but the fiber network remains structurally intact (Fig. 1A and supplementary movie 1, and ⁵) and functionally intact (conductivity is retained after SDS/EDTA extraction, Fig. 4B/D, and ⁷). This chemical resistance could be aided by protein disulfide bridges, which could be abundant as suggested by the high S content (Fig. 4F). The two observed organic S fragments ($C_2S_2^-$ and $C_2S_2H^+$) could hence come from the disulfide bridges.”

o If disulfide bonds are so important for structure and conductivity, the chemical reduction of the fiber sheath iron cyanide would have affected these fiber properties. Yet, the authors indicate that, “reduced fiber sheaths showed a 2.1 ± 0.5 (N = 11) higher conductance than oxidized fiber sheaths” (line 248, main text).

It is unlikely that the iron cyanide reduction would break disulfide bridges in proteins, which would irreversibly change protein structure. However, both our Raman and conduction redox experiments suggest that the fiber sheaths can be reversibly oxidized and reduced (Fig. 5A/B, see also method description of these experiments).

- **Discussion: Overstatements abound. I note some examples:**

We toned down the wording in the discussion at specific points.

o Line 302: “Our results demonstrate that conduction in cable bacteria occurs through proteins with Ni-dependent cofactors”. Demonstrate → indicate

o Line 332: “At this moment, the exact mechanism of conduction remains unclear, but our results demonstrate that the novel Ni-cofactor is an essential component”. Demonstrate → indicate

o Line 334: “... we also detected substantial signals from aromatic amino acids in the fiber proteins, one possibility is that conduction is based on electron transfer between S-ligated Ni-groups assisted by bridging aromatic groups in nearby aromatic amino acids”.

The aromatic amino acid discussion was toned down substantially (see response to comment 3 of reviewer 3).

☒ What does “substantial” mean in this context? (Note that there are many aromatic-rich proteins that are not conductive.)

We answer this in detail at comment 3 of reviewer 3.

☒ A model of hybrid conduction involving S-ligated Ni and aromatic residues requires knowledge of the distribution of metal and aromatic clusters. The LEXRF maps show metal clusters but the organization of aromatics is not known. Furthermore, the aromatic clusters would have to bridge the large distance between the metal clusters.

Valid point. We have substantially changed the discussion to accommodate this suggestion. Electron transport in metalloproteins critically depends on the distance between metal centers. We have therefore added estimates of the interatomic Ni distances in the conductive core of the fibers (1.4 to 1.9 nm). This is well within the range commonly observed for electron transport in metalloproteins. We also included comparison with the recently discovered cytochrome nanowires in *Geobacter* discussing electron transfer times and conductivity (see previous detailed response above).

☒ How does Cu fit in this model? (Note the Cu clusters).

It does not fit in the model as Cu is most likely mainly from an impurity during LEXRF analysis as discussed in the text and above in the rebuttal. The 3 other methods used to analyze metals show low abundance of Cu.

o Line 339: “... promising gateway for new technology, and creates the prospect of bio-electronic devices with new functionality that integrate proteins as new class of electronic materials.” Big overstatement at the end of the discussion when they do not even know whether this is an organized assembly and what proteins make it. It also surprises considering that technologies based on the *Geobacter pili* have been realized already.

We have removed the whole short paragraph including this line at the end of the discussion.

Reviewer #3 (Remarks to the Author):

Boschker and coauthors describe a set of analytical experiments to determine the chemical features of cable bacteria sheaths that support long-range electronic conductivity. In my opinion, their conclusions are well supported and this manuscript represents a significant advance to understanding the structural and chemical underpinnings of the most conductive and longest microbial structures characterized to date. My comments are relatively minor and I imagine they could be readily addressed by the authors.

We thank the reviewer for the excellent comments that certainly improved the manuscript and general support for the study.

1. First, a general comment about softening the language used throughout the manuscript. The area of research focusing on understanding the chemical and structural features that allow for long-range

electronic conductivity in microbial structures has undergone an upheaval in the past several years, with new structures and new understanding calling into question many long-held beliefs about what is or is not possible and providing new insights into how anaerobes respire. The understanding of why these cable bacteria sheaths are so electrically conductive has been and will continue to be a challenge. Even with this manuscript, which is a significant advancement towards understanding the working principles of these sheaths, the field still lacks a molecular and genetic basis of these structures. Lacking these more conclusive pieces of evidence, I highly recommend softening the language in the paper to present results as “suggesting” or “consistent with” or “providing further support for” a hypothesized model. For example, the usage of “indicate/s/d/ing” in lines 110, 124, 180, 186, 257, 365, 394 and “confirm/s/ed/ing” in lines 107, 173, 205, 210, 255, 259 should be softened. Many of the experiments in this work are indirect and subject to artifacts. To keep the presentation of results objective, I suggest a rewording of the implications of these outcomes. The data are consistent with a model, but do not prove it conclusively, and should be presented as such.

We agree with the reviewer and have softened the language at the suggested positions where appropriate (changes are marked in the revised manuscript text). We did however not change “indicated” at line 110 (line 111 in revised manuscript) as AFM-IR spectra leave little doubt and fully agree with a protein and polysaccharide composition. Also, “confirmed” at line 173 (line 178 in revised manuscript) was kept as the Raman spectra of the fiber sheaths do show that the lower bands are associated with the cell envelope (as originally inferred from distribution of Raman signals in intact filaments). An additional change was made in line 220 (indicated → suggested).

2. Line 119-126 ignores recent results suggesting a role other than conductivity for Pila aromatic species. Aside from the problematic lack of an experimentally determined atomistic structure that supports a chemical and structural basis for electronic conductivity in *G. sulfurreducens* Pila, Yalcin et al Nat Chem Biol 16, 1136-1142, 2020 show that mutations to aromatic amino acids affect cytochrome secretion which in turn affects the proportion of more or less conductive cytochrome wires. So while the speculation on lines 333-336 that aromatic amino acids may bridge metal-sulfide cofactor sites is not off base, the rationale that aromatic amino acid content alone can be responsible for conductivity as suggested in 119-126 needs to be better contextualized.

The Yalcin et al 2020 paper was published after we first submitted and has now been added as a reference at line 183. It is further referenced in a comparison between the fibers in cable bacteria and the structure and conduction of the OmcZ nanowires in the extended discussion. We also rewrote the part about the possible role of aromatic amino acids in line with the recent evidence that the *Geobacter* appendages are not pili as long thought but instead are made of multi-heme cytochromes.

See also our response to comment 3 on aromatic amino acids by this reviewer. The discussion was substantially rewritten in the revised version to better include the recent work on *Shewanella* and *Geobacter* nanowires (starting from line 369).

3. Related to the previous point, the appearance of aromatic signal in IR and ToF-SIMS does not necessarily mean aromatic amino acids are more abundant than in a typical protein. Comparisons to control protein signals by the same techniques, either by additional experiments or comparison to

literature data would help contextualize the assessment that the signal constitutes an “aromatic rich” protein.

This is a good point raised by the reviewer. We had another detailed look into this matter, and as a result, we no longer support the claim that the fibers may be rich in aromatic amino acids.

The AFM-IR spectra (Fig. 1B) do show an aromatic C-H band at 3056 1/cm that is rather unusual for a material that mainly consists of protein and polysaccharide. AFM-IR has been widely used to study IR spectra of individual proteins for which the amino acid composition is available. However, even after an extensive review of the current literature, we were unable to find any AFM-IR protein study that covers the C-H range of the IR spectrum. All studies on single proteins so far have focused on the Amide I and II bands in the 1500-1600 1/cm region as these provide information on protein secondary structure. Also, standard bulk FTIR spectra from proteins typically do not show an aromatic C-H band (eg. Barth 2004), but this could be method related.

In addition, specific fragment ions from all three aromatic amino acids were detected by ToF-SIMS, but it is difficult to put their counts in a quantitative perspective. ToF-SIMS has been used extensively to differentiate between proteins based on a range of specific fragments from amino acids (eg. Baugh et al., 2010; Lebec et al., 2014). Statistical techniques like principle component analysis are used to identify the proteins, but we have not been able to find a study where ToF-SIMS was used to quantify amino acid composition. We performed a ToF-SIMS method test using two proteins with different amino acid compositions (BSA and gelatin), which suggests that response factors for individual amino acids may differ substantially between proteins possibly as the result of matrix effects (data not shown).

It should be noted that the claim that the fibers may be rich in aromatic amino acids was not essential for the main conclusions of the manuscript. The revised manuscript as before focusses on the role of the Ni/S-group the electron conduction and the identity, structure and composition of the proteins involved is treated in the discussion as important suggestions for further research.

Changes made in the text:

Line 115: Now reads “A band at 3056 cm⁻¹ likely represents aromatic C-H stretching for instance from aromatic amino acids.”

Line 149: deleted “is rich in aromatic amino acids”

Rewrote the discussion on this point, which now reads

“The fast electron tunneling times of 50-90 nsec suggests that the conductive fiber core must possess a supramolecular organization that supports highly efficient electron transport between Ni centers. This could involve particular configurations of aromatic amino acids, or maybe an unknown prosthetic group involved in Ni-ligation. Future studies should further elucidate the genetic and molecular basis of the highly efficient long range electron transport in cable bacteria.” (line 419).

4. The core-shell fiber model seems reasonably supported by evidence, but it is unclear how it works in a biological context. If the conductive elements are shielded from the surrounding environment, how do the Desulfobulbaceae dump electrons into the structures in the anoxic zone, how does molecular oxygen access the reducing potential in the oxic zone, and how do other anaerobes access

the oxidative potential of the cables in the anoxic zone, as has been shown elsewhere? The selective insulating properties of the shield layer should be clarified.

Good point. We currently don't know how exactly this works. In the genomic paper of Kjeldsen et al 2019, it is argued that periplasmic cytochromes as found in the genomes of cable bacteria may be involved, but there is however no direct proof for this suggestion. Cytochromes are however removed during the extraction of the fiber sheaths, which suggests that this shuttle pathway is no longer present in the extracted fiber sheaths.

We now discuss this in the revised manuscript at line 313:

"However, the insulating outer layer also provokes questions on how electrons generated during electrogenic sulfur oxidation are uploaded onto or downloaded from the conductive core, i.e., how these electrons pass through the insulating layer of the fibers to reach the inner conductive core. Cable bacteria must contain a mechanism for this electron transport to and from the conductive fiber core, which may involve periplasmic cytochromes¹² that are present in intact bacteria, but are removed during fiber sheath extraction (Fig. 2A and 7)."

5. Figure 4D is confusingly presented. Do the lines represent before and after EDTA treatment currents of individual sheaths? How are the sheaths arranged left to right, by decreasing current reading? In my opinion the information would be more effectively presented as a comparison of the means (+/-standard error) of the "before" samples and "after" samples. Alternatively, it could be presented as the mean change in current in the "before" samples and "after" samples. Either way, the current presentation makes it difficult to interpret.

Figure 4D has been changed to a box-whisker plot in agreement with this comment and the Nature Comm. formatting instructions.

6. The statistical methods used are unfamiliar to me. Could the authors explain why the Wilcoxon test was used instead of the more commonly used Student's t-test?

The Wilcoxon test is the non-parametric version of the Student's t-test. However, it is also called the Wilcoxon signed-rank test, the Mann-Whitney test or the Mann-Whitney-Wilcoxon test, which may explain the confusion.

We now refer to it in the legend of Figure 4 as "The Mann-Whitney-Wilcoxon test was used to test for significance using the R-function 'wilcox.test'.", line 509.

7. Lastly, is there a genetic basis for thinking that Ni-S metalloenzymes are made by Desulfobulbaceae? If the authors could establish that finding Ni-S containing proteins in these organisms is expected from genome analysis, that would bolster the molecular model they are building with this work.

We certainly had a look in the published genomes of cable bacteria but could not readily identify possible novel S-ligated Ni proteins. Of the known S-ligated Ni proteins, Ni-SOD is not present and cable bacteria use another type of SOD. The ACS gene from the Ljundahl-Wood pathway is present in cable bacteria, but this contains a combined Fe and Ni center plus a couple of FeS-clusters. There is

however very little Fe associated with the Ni-protein based on the ToF-SIMS depth profiles from fiber sheaths. Also, there are several Ni uptake and binding proteins present in the genome, but these typically do not show S-ligation.

It should be noted that the unknown conductive fiber Ni-protein may well have a not yet described Ni-binding motive as it performs a very different function than the known Ni-proteins, which are primarily involved in the metabolism of gasses. This means that it will be difficult to detect the Ni-protein gene based solely on genome sequences. Approximately 60% of the open reading frames in the available cable bacteria genomes show no close relationship with genes with known functions.

The data presented in the manuscript does not provide much scope for discussing the genes and proteins that might be involved in fiber electron transport in any detail. We however added a statement to the discussion that “Future studies should further elucidate the genetic and molecular basis of the highly efficient long range electron transport in cable bacteria.”, line 423.

Otherwise, the manuscript is excellent and represents a substantial advance in the understanding of the structures supporting long-range electronic conductivity in these exceptional bacteria.

REVIEWERS' COMMENTS

Reviewer #2 (Remarks to the Author):

The revised manuscript is a significant improvement compared to the earlier version I reviewed. The authors did a great job at toning down the conclusions, at considering alternative explanations and at carefully building the arguments in support of a core-shell model. I only have some minor suggestions for the author's consideration that are aimed at improving the accessibility of the material to the journal's broad audience.

1. Readers may be unfamiliar with the terminology used to describe the fiber sheaths and their regions/components. This can be alleviated by including labels in Fig. 1A. For example, the B&W image can indicate where the cell junctions (constricted regions of the sheath) versus central cell regions are. They can use the image reconstructions (in blue) to highlight the parallel arrangement of individual fibers and indicate where the basal sheaths are (they also need to explain what they consider "rings of sheaths"). These are all terms that are difficult to understand without prior knowledge of the specific literature or a deep understanding of terms in the Introduction.

2. The AFM-IR spectroscopic studies suggest that the sheaths are made of protein (potential signatures could be from aromatic amino acids) and polysaccharide (with acidic sugars). This is all clearly stated.

a. Spectral intensity of the protein amide peak is highest at the cell junctions. They indicate that this is "likely due to presence of the cartwheel structure that interconnects fibers⁵". Can they conclude that the cartwheel structure is made of protein? Some explanation is needed here, otherwise the statement feels orphan.

b. They revised the last paragraph in this section (line 129) in a rather convoluted way. All that I suggested was to clearly state that the results argue about pilins and or pilin assemblies (pili) being part of the structure, as it was suggested elsewhere.

c. Emphasizing that these proteins are unlikely to be pili is important because it shows that there is something new here and potentially involved in the remarkable conductivity of these fibers. They can emphasize this in the concluding paragraph (line 149: "(ii) that the fibers consist of protein UNLIKELY TO BE PILI").

DETAILED RESPONSE TO THE REVIEWER COMMENTS (the response is marked in blue font)

Reviewer #2 (Remarks to the Author):

The revised manuscript is a significant improvement compared to the earlier version I reviewed. The authors did a great job at toning down the conclusions, at considering alternative explanations and at carefully building the arguments in support of a core-shell model. I only have some minor suggestions for the author's consideration that are aimed at improving the accessibility of the material to the journal's broad audience.

We thank the reviewer for the positive evaluation of the revised manuscript.

1. Readers may be unfamiliar with the terminology used to describe the fiber sheaths and their regions/components. This can be alleviated by including labels in Fig. 1A. For example, the B&W image can indicate where the cell junctions (constricted regions of the sheath) versus central cell regions are. They can use the image reconstructions (in blue) to highlight the parallel arrangement of individual fibers and indicate where the basal sheaths are (they also need to explain what they consider "rings of sheaths"). These are all terms that are difficult to understand without prior knowledge of the specific literature or a deep understanding of terms in the Introduction.

As suggested, we labeled the junctions and central cell area in Fig. 1A. And reworded the sentence to exclude the ring of fibers "demonstrating that the regularly spaced fibers appear to be held together by a basal sheath."

2. The AFM-IR spectroscopic studies suggest that the sheaths are made of protein (potential signatures could be from aromatic amino acids) and polysaccharide (with acidic sugars). This is all clearly stated.

a. Spectral intensity of the protein amide peak is highest at the cell junctions. They indicate that this is "likely due to presence of the cartwheel structure that interconnects fibers". Can they conclude that the cartwheel structure is made of protein? Some explanation is needed here, otherwise the statement feels orphan.

We focused the manuscript on the composition of the fibers at the central cell area and mostly refrained from drawing conclusions on the composition of the card wheel at the junctions in the manuscript. In our samples the cart wheel is always buried underneath a layer of fibers that is continues across the cell junction. Other approaches probably involving making cross sections would be needed to with certainty study the composition of the cartwheel without interference from the fibers.

However, the AFM-IR spectra at the junctions were very similar to the cell area suggesting a similar composition of mainly of protein and polysaccharide, but signals were substantially higher at the junctions. This indeed may suggest that the higher signals are due to the cartwheel at the junction and therefore that the card wheel also exist mainly of protein and polysaccharide.

We have added that the cartwheel most likely also is made of protein and polysaccharide: "Cell junctions give similar AFM-IR spectra as central cell areas, although signals are higher

in the junctions (Fig. 1B and 1C), likely due to presence of the cartwheel structure that interconnects fibers⁵ and also seems to be made of protein and polysaccharide.”

b. They revised the last paragraph in this section (line 129) in a rather convoluted way. All that I suggested was to clearly state that the results argue about pilins and or pilin assemblies (pili) being part of the structure, as it was suggested elsewhere.

The exact position of the Amide I peak in the AFM-IR spectra is not in agreement with a large amount of alpha-helices as would be expected when pilin protein would be a major component of the fiber sheath. However, we also calculated that the fibers are only ca. 25% of the total fiber sheath and realized that we may be looking at much more proteins than just from the fibers. It is therefore not possible to draw strong conclusions on the secondary structure of the fiber proteins.

We have simplified the end of the paragraph to:

“The position of the Amide I peak at 1643 cm^{-1} suggests however that the protein secondary structure is mainly disordered, and not of the α -helix type¹⁰, which hence speaks against an abundance of α -helix rich pilin proteins. We estimate that the fibers themselves represent 25% of the total mass of the fiber sheath (see Supplementary Note 3), and so other proteins outside of the fibers can contribute to the Amide I peak. While it is unlikely that pilin protein is present in high concentrations in the fibers, its complete absence requires further confirmation.”

c. Emphasizing that these proteins are unlikely to be pili is important because it shows that there is something new here and potentially involved in the remarkable conductivity of these fibers. They can emphasize this in the concluding paragraph (line 149: “(ii) that the fibers consist of protein UNLIKELY TO BE PILI”).

The complete absence of pilin proteins needs further confirmation, and hence we believe that it would be a too strong conclusion. Yet, we have now given the likely absence of pilin protein more emphasis in the preceding section (see our response to comment b).

“While it is unlikely that pilin protein is present in high concentrations in the fibers, its complete absence requires further confirmation.”